# Interplay of adherens junctions and matrix proteolysis determines the invasive pattern and growth of squamous cell carcinoma

Takuya Kato[1,2†], Robert P Jenkins[1*†], Stefanie Derzsi[1,3], Melda Tozluoglu[4], Antonio Rullan[1,5], Steven Hooper[1], Raphaël AG Chaleil[4], Holly Joyce[1], Xiao Fu[1,4], Selvam Thavaraj[6], Paul A Bates[4], Erik Sahai[1*]

[1]Tumour Cell Biology Laboratory, The Francis Crick Institute, London, United Kingdom; [2]Department of Pathology, Kitasato University, Sagamihara, Japan; [3]Hoffman La-Roche, Basel, Switzerland; [4]Biomolecular Modelling Laboratory, The Francis Crick Institute, London, United Kingdom; [5]Institute of Cancer Research, London, United Kingdom; [6]Centre for Oral, Clinical and Translational Sciences, King's College London, London, United Kingdom

**\*For correspondence:**
robert.jenkins@crick.ac.uk (RPJ);
erik.sahai@crick.ac.uk (ES)

†These authors contributed equally to this work

**Abstract** Cancers, such as squamous cell carcinoma, frequently invade as multicellular units. However, these invading units can be organised in a variety of ways, ranging from thin discontinuous strands to thick 'pushing' collectives. Here we employ an integrated experimental and computational approach to identify the factors that determine the mode of collective cancer cell invasion. We find that matrix proteolysis is linked to the formation of wide strands but has little effect on the maximum extent of invasion. Cell-cell junctions also favour wide strands, but our analysis also reveals a requirement for cell-cell junctions for efficient invasion in response to uniform directional cues. Unexpectedly, the ability to generate wide invasive strands is coupled to the ability to grow effectively when surrounded by extracellular matrix in three-dimensional assays. Combinatorial perturbation of both matrix proteolysis and cell-cell adhesion demonstrates that the most aggressive cancer behaviour, both in terms of invasion and growth, is achieved at high levels of cell-cell adhesion and high levels of proteolysis. Contrary to expectation, cells with canonical mesenchymal traits – no cell-cell junctions and high proteolysis – exhibit reduced growth and lymph node metastasis. Thus, we conclude that the ability of squamous cell carcinoma cells to invade effectively is also linked to their ability to generate space for proliferation in confined contexts. These data provide an explanation for the apparent advantage of retaining cell-cell junctions in squamous cell carcinomas.

## Editor's evaluation

This article constitutes a carefully written and highly impactful study that agrees with recent paradigm-shifting studies (e.g., doi: 10.1126/scitranslmed.abn7571 and doi: 10.1242/jcs.259275) suggesting that collective cell migration is the most efficacious way for epithelial cells to metastasize. The study uses mathematical modeling and experimental 3D approaches to demonstrate that cells necessitate space to both proliferate and invade as collective thick "pushing" strands. Importantly, extracellular matrix patterning provides uniform directional cues that harness adherens junctions and facilitate the collective thick strands of cells to 'push' and effectively travel through 3D microenvironmental settings. The study breaks new ground by incorporating cancer-associated fibroblasts and concludes that the pushing fronts are allied to extracellular matrix proteolysis and

strong cancer cell-cell adherens junctions with diminished dependence on stromal cells (e.g., cancer-associated fibroblasts).

## Introduction

Tumours exhibit a variety of histological patterns that inform pathological diagnosis and that are frequently linked to prognosis (*Dive et al., 2014*). This link with outcome suggests that the mechanisms specifying histological pattern are related to tumour malignancy. This may be due to some coupling between how cancer cells invade and their ability to proliferate. Epithelial cancer cells, including squamous cell carcinoma (SCC), frequently invade in collective units (*Friedl and Gilmour, 2009*; *Khalil et al., 2017*; *Wang et al., 2016*). The importance of collective invasion is underscored by several recent studies showing that collective seeding of metastases is more efficient than single cell seeding (*Cheung et al., 2016*; *Fischer et al., 2015*; *Khalil et al., 2017*; *Padmanaban et al., 2019*). Despite the prevalence and importance of collective patterns of cancer cell invasion, it remains less well understood than single cell forms of invasion. Collectively invading cancer strands can be organised in a variety of different ways, from single file strands that characterise invasive lobular breast cancer and diffuse gastric cancer to broad cohesive units found in basal cell carcinoma (*Boelens et al., 2016*; *Carneiro et al., 2004*; *Friedl et al., 2012*; *Pandya et al., 2017*). Histological analysis indicates that even within a single disease type there is considerable heterogeneity in the pattern of invasion; for example, both broad 'pushing' and strand-like infiltrative invasion can be observed in SCC (*Dissanayaka et al., 2012*). In this study, we set out to explore the key parameters that determine the pattern of collective invasion using a combination of computational and experimental approaches.

Several parameters might be expected to modulate tumour histology and, more specifically, collective cancer cell invasion. The ability of cancer cells to adhere to each other through cadherin-mediated junctions is linked to their organisation into tightly packed clusters. E-cadherin/*CDH1* and, to a lesser extent, P-cadherin/*CDH3* are the predominant cadherins in mucosal squamous cell carcinomas (SCC or muSCC specifically for mucosal SCC; *Nieman et al., 1999*) that typically do not undergo a clear epithelial to mesenchymal transition (EMT). These cadherins are coupled to the actin cytoskeleton via a complex containing α-catenin and β-catenin (*Nelson et al., 2013*). Cell adhesion to the extracellular matrix (ECM) is also critical for cell migration and invasion in many contexts (*Cooper and Giancotti, 2019*; *Hamidi and Ivaska, 2018*). This is primarily mediated by integrin receptors (*Hamidi and Ivaska, 2018*; *Janes and Watt, 2006*), with *ITGB1* particularly highly expressed in SCC (*Janes and Watt, 2006*). The ECM presents a barrier to migration if the gaps between fibres are smaller than 3–5 µm (*Wolf et al., 2013*; *Wolf et al., 2009*). The dermal ECM underlying SCC lesions is predominantly composed of type I collagen (*Watt and Fujiwara, 2011*), and numerous studies have demonstrated that MMP14/MT-1MMP is the critical protease for degrading this type of matrix (*Castro-Castro et al., 2016*; *Gifford and Itoh, 2019*). The ECM can also be physically moved by forces generated by the contractile cytoskeleton (*Mohammadi and Sahai, 2018*; *Wolf et al., 2013*). In many cases, stromal cells are the major source of both matrix proteolytic and force-mediated matrix remodelling in tumours (*Conklin and Keely, 2012*; *Kalluri and Zeisberg, 2006*). Cancer-associated fibroblasts (CAFs, sometimes referred to as stromal fibroblasts) can promote the invasion of SCC by providing these functions and are frequently observed leading the migration of cancer cells that retain epithelial characteristics (*Gaggioli et al., 2007*).

Understanding the relative contributions of the multiple parameters outlined above to cell invasion is a complex multi-dimensional problem with non-linear relationships between parameters and migratory capability. This complexity means that developing a holistic and predictive framework for collective cancer cell invasion using empirical methods only is challenging. For this reason, several studies have sought to utilise computational models. Many different types of model have been used including those based on evolutionary game theory (*Basanta et al., 2008*; *Swierniak and Krzeslak, 2013*), Bayesian networks (*Katz et al., 2011*), differential equations (*Gerisch and Chaplain, 2008*; *Peng et al., 2017*; *Weekes et al., 2014*), agent-based models including cellular automata (*Alarcón et al., 2003*; *Bull et al., 2020*; *Fiore et al., 2020*; *Gralka and Hallatschek, 2019*; *Karolak et al., 2019*; *Norton et al., 2017*; *Talkenberger et al., 2017*) and hybrids of the above (*Anderson, 2005*; *Anderson et al., 2006*; *Osborne et al., 2010*). Cellular Potts modelling (*Cickovski et al., 2007*; *Graner and Glazier, 1992*; *Hallou et al., 2017*; *Pramanik et al., 2021*; *Scianna et al., 2013*; *Shirinifard et al.,*

*2009*; *Szabó and Merks, 2013*; *Turner and Sherratt, 2002*) is a flexible approach that uses voxels to represent different parts of cells or their environment. Changes in the properties associated with each voxel are determined at each time step using principles of probabilistic energy minimisation. The behaviour of the model therefore emerges from iterative application of rules that describes the relative favourability of different events or changes. Here we combine a Potts modelling with extensive experimentation to unpick the determinants of the mode of collective cancer cell invasion and their linkage to cancer cell growth, both in vitro and in vivo.

## Results

### Diverse modes of collective invasion within individual SCC

We began by surveying the diversity of invasive pattern in muSCC. *Figure 1a* shows considerable diversity in the nature of collective invasion. Furthermore, it illustrates how initial invasion involves cells moving from the epithelial layer into the lamina propria (often termed epidermis and dermis, respectively, in cutaneous skin). Following the invasion into the dermis, cancer cells become surrounded on all sides by ECM. Interestingly, different patterns of invasion were observed in different regions of the same tumour (*Figure 1a*). These ranged from broad 'pushing' invasive masses of cells (box I), thinner strands of cells (box II), to single-cell width strands and apparently isolated single cells (box III - although this could not be definitively determined from single H&E (Hematoxylin and Eosin) sections). Quantitative analysis of the number of cell neighbours provided a more objective metric of invasion type, with high neighbour numbers (typically 4–7) indicating broad invasion patterns and low neighbour numbers (2 or 3) indicating thin strand-like invasion, respectively (*Figure 1b*). Similar patterns were observed in other muSCC biopsies with different strand thickness apparent (*Figure 1—figure supplement 1a–f*). Analysis of neighbour number suggested that strand thickness does not fall into distinct categories, with neighbour number varying continuously between 1 and 9.

To gain insight into the dynamics of SCC invasion, we performed time-lapse imaging of primary patient explants. Small pieces of tissue, roughly 1 mm$^3$ in size, were embedded in a collagen-rich matrix and observed by time-lapse microscopy. Similar to the diversity observed in histological sections, this revealed a variety of behaviours, including single-cell 'follow the leader' migration through to large 'dome-like' multi-cellular invasion, even in samples from a single patient (*Figure 1c*). Cell tracking revealed that, in the larger invading structures, there was movement both in the direction of invasion and retrograde back to the main bulk of the explant. The diversity of collective invasion phenotype within a single tumour suggests that the type of collective invasion is not irreversibly determined by early events in the history of the tumour but can be influenced by variations in cell state that may occur later in tumorigenesis or local environmental differences.

### Generation of an agent-based model of collective cancer invasion

To explore the possible variables responsible for the different collective invasive behaviours observed, we set up both experimental and computational models. Two different experimental settings were implemented. First, an 'organotypic' invasion assay in which the SCC cells are cultured as a layer on top of a collagen-rich matrix and exposed to a gas-liquid interface. This recapitulates the early invasion of disease from the epidermis into the dermis (as in the top region of *Figure 1a*). Second, a 'spheroid' assay was used in which the SCC cells are encapsulated in a collagen-rich matrix, mimicking the more confined environment of disease that has already penetrated into the dermis (as in the bottom region of *Figure 1a*). Alongside these two experimental contexts, we developed a cellular Potts model that incorporated both SCC cells and stromal fibroblasts. The interaction of cancer cells with ECM and fibroblasts during invasion has been extensively modelled computationally in recent years (*Arduino and Preziosi, 2015*; *Kim et al., 2015*; *Kumar et al., 2016*; *Norton et al., 2018*; *Pally et al., 2019*; *Sfakianakis et al., 2020*). In our three-dimensional (3D) model, the voxel size was such that cells typically consisted of 400–800 (~8$^3$) voxels. Cell invasion could occur by a cell moving a voxel to a position that was previously occupied by matrix. To determine whether such a change might be favourable, the model included parameters that we anticipated would influence cancer cell invasion, including cancer cell–cancer cell adhesion, cancer cell–matrix adhesion, cancer cell–fibroblast adhesion, fibroblast–matrix adhesion, cell intrinsic motility, matrix displacement, and matrix proteolysis. The relative influence of these parameters on changes in the position of voxels that defined a cell

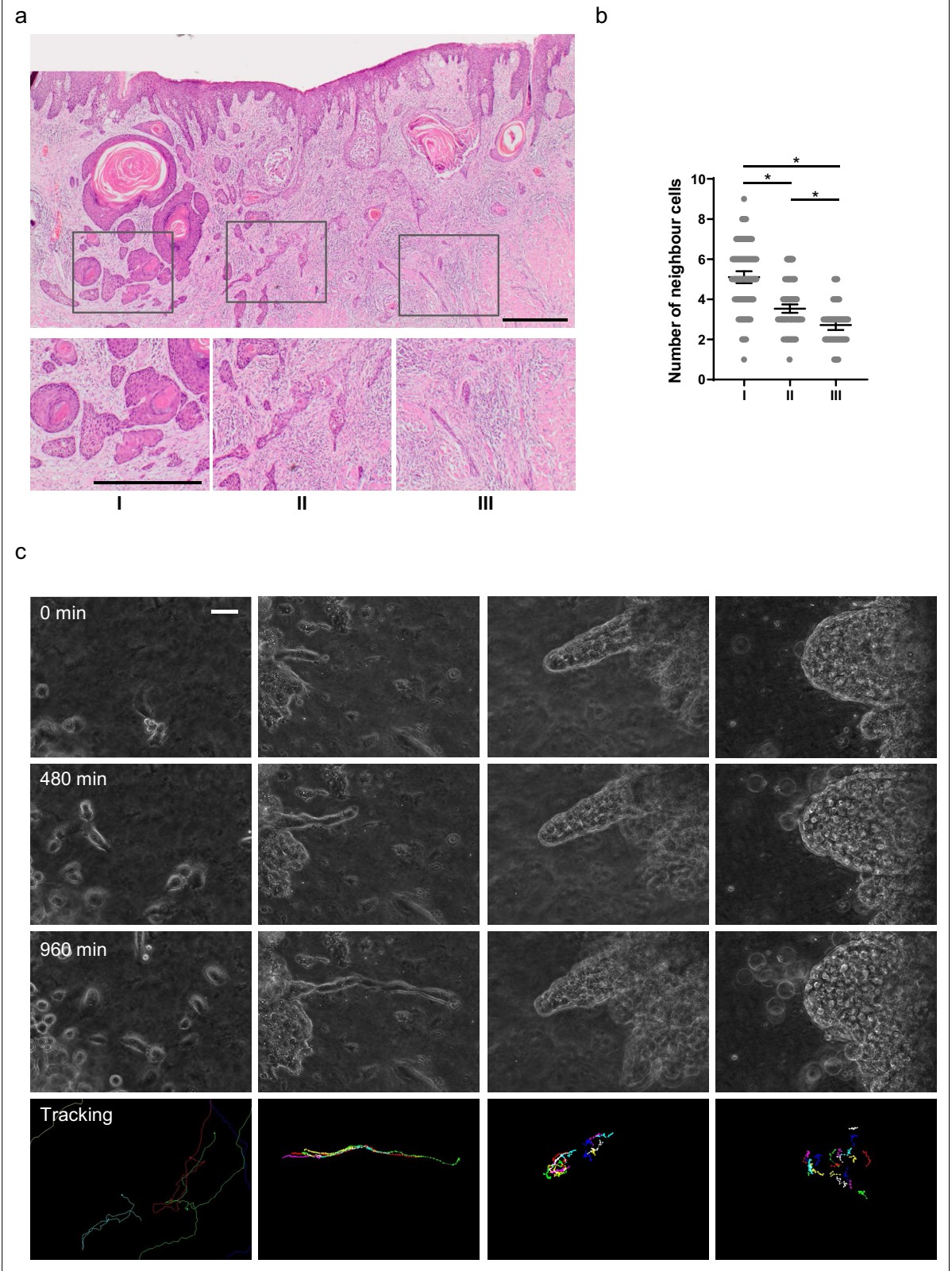

**Figure 1.** Diversity of collective invasion in squamous cell carcinoma (SCC). (**a**) Images show a human invasive head and neck SCC stained with haematoxylin and eosin. Inset regions show different patterns of collective invasion: I – large rounded clusters, II – intermediate clusters, III – elongated strands only one or two cells wide. (**b**) Plot shows the mean number of cancer cell neighbours for each cell within invasive strands with the morphologies exemplified in panel (a) I, II, and III. One-way ANOVA with post-hoc multiple comparisons was performed. 95% confidence intervals are shown, one dot

*Figure 1 continued on next page*

*Figure 1 continued*

represents one cell analysed. (**c**) Images show phase contrast microscopy of a human oral SCC invading into a collagen/Matrigel mixture. Scale bar is 100 μm. Lower panels show manual tracking of individual cells within the clusters.

The online version of this article includes the following video, source data, and figure supplement(s) for figure 1:

**Source data 1.** Number of neighbouring cells for each cell within invasive strands in oral SCC tissue.

**Figure supplement 1.** Diversity of collective invasion in squamous cell carcinoma.

**Figure supplement 1—source data 1.** Number of neighbouring cells for each cell within invasive strands in head and neck SCC tissue.

**Figure 1—video 1.** Single cell invasion: squamous cell carcinoma cells invading as single cells.
https://elifesciences.org/articles/76520/figures#fig1video1

**Figure 1—video 2.** Single file strand invasion: squamous cell carcinoma cells invading as single file strand.
https://elifesciences.org/articles/76520/figures#fig1video2

**Figure 1—video 3.** Thick strand invasion: squamous cell carcinoma cells invading as thick strand.
https://elifesciences.org/articles/76520/figures#fig1video3

**Figure 1—video 4.** Pushing front invasion: squamous cell carcinoma cells invading as broad pushing invasive mass.
https://elifesciences.org/articles/76520/figures#fig1video4

between time-steps was determined along energy minimisation principles, with penalties of differing magnitudes for unfavourable changes in any single parameter (*Figure 2a* and Appendix 1 in *Supplementary file 1*).

Experimental analysis using A431 SCC cells demonstrated that effective invasion required the addition of CAFs and, in both cases, the invasion was almost entirely collective (*Figure 2b* panels iii and vi). Careful parameterisation was performed, including analysis of the relative adhesive properties of the different cells to each other and the collagen-rich matrix used in our assays (*Figure 2—figure supplement 1a* and *Table 1* and Appendix 2 – table 1 in *Supplementary file 1*). This enabled the in silico replication of the fibroblast-dependent invasion observed in both organotypic and spheroid assays (*Figure 2b*). In line with previous experimental reports (*Gaggioli et al., 2007*), the extent of increased invasion scaled with the number of fibroblasts (*Figure 2—figure supplement 1a*).

## In silico generation of diverse collective invasion behaviours

Having established an in silico model, we then explored parameter space to investigate if different patterns of invasion could be generated by varying the combinations of input parameters. To quantitatively capture the range of invasive behaviours, a range of output metrics were collected, including total invasive extent, maximal invasion, number of cell neighbours, and cell proliferation (*Figure 2—figure supplement 1b* and c). The tapering metric recorded how the number of immediately neighbouring cells varied with the position of the cells in the invasive strand (cells were considered invasive if they had moved beyond the starting position of the interface between cancer cells and the matrix), whereas the strand width simply reflected the average width. A uniformly low neighbour number would indicate a long thin strand (*Figure 2—figure supplement 1cI*), a decreasing number of neighbours with increasing invasion would indicate a tapering strand (*Figure 2—figure supplement 1cII*), while a higher number of neighbours would suggest a bulkier form of collective invasion (*Figure 2—figure supplement 1cIII*). A critical function of fibroblasts is to generate permissive tracks for cancer cells to subsequently utilise. To mimic this without the variability generated by the somewhat stochastic behaviour of fibroblasts, we additionally ran simulations with a narrow track that could be permissive for invasion but no fibroblasts. This confirmed that cancer cells were able to exploit permissive tracks in the ECM (*Figure 2—figure supplement 1d*). Invasion in this context, termed track invasion score, was quantified based on the extent of matrix remodelling by invading cancer cells with weighting for the distance invaded (*Figure 2—figure supplement 1e*).

The outputs of the model in the presence of CAFs were analysed in two ways: using principal component analysis (PCA) and visual inspection (*Figure 2c and d*). PCA revealed a wide and continuous spread of invasion patterns, with the first two dimensions of the PCA accounting for 75% (organotypic) and 65% (spheroid) of the variation (Appendix 3 – table 1 in *Supplementary file 1*). Notably, there was no indication of discrete sub-classes of invasive pattern, suggesting a continuous spectrum of invasive behaviours. The continuous spectrum implied by PCA was in line with the range of invasive

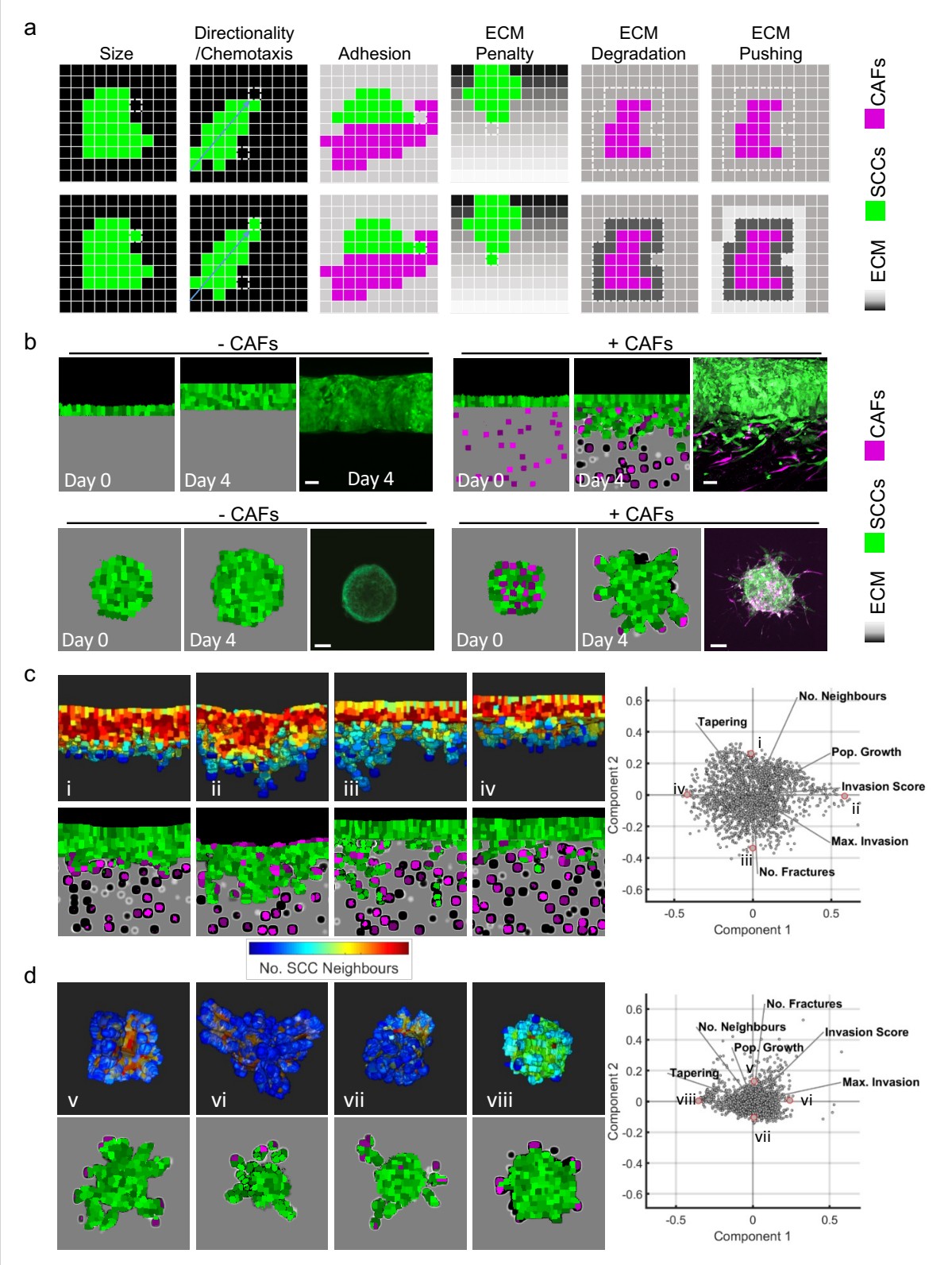

**Figure 2.** Agent-based modelling recapitulates diversity of collective invasion patterns. (**a**) Images show the key steps and principles driving the agent-based model. At each time step, voxels are updated, and this can include growth – matrix replaced by cell, directional movement – cell voxel transposition, cell-cell adhesion – increase of voxels at the interface between cells, and extracellular matrix (ECM) remodelling or degradation – change in the 'quantity' of matrix in a voxel. Cancer cells are represented in green, fibroblasts in magenta, and matrix in greyscale. (**b**) Images show model

*Figure 2 continued on next page*

*Figure 2 continued*

outputs (panel columns 1, 2, 4, and 5) next to experimental data when fibroblasts are either absent or present in organotypic models (upper panels) or spheroids (lower panels). Cancer cells are green, fibroblasts are magenta. Scale bar = 100 µm. (**c** and **d**) Diverse patterns of collective invasion are shown in organotypic (**c**) and spheroid (**d**) models. Principal component analysis plot shows the metrics derived from over 2000 simulations in the presence of fibroblasts covering variation in cancer cell proteolysis, cancer cell–matrix adhesion, and cancer cell–cancer cell adhesion. The additional lines indicate how different metrics contribute to the first two components. The model runs corresponding to the exemplar images are indicated with roman numerals with three-dimensional images coloured according to mean number of squamous cell carcinoma (SCC) neighbours (blue low, red high). CAFs, cancer-associated fibroblasts.

The online version of this article includes the following figure supplement(s) for figure 2:

**Figure supplement 1.** Agent-based modelling recapitulates diversity of collective invasion patterns.

strand geometries observed in clinical muSCC samples (*Figure 1*). We additionally generated visual outputs of the model runs that lay at the edges of the PCA. This revealed diverse patterns of invasion, ranging from large rounded multicellular strands to single cells breaking off the main mass of tumour cells. The diversity in collective invasion observed in the presence of fibroblasts was in contrast to behaviours observed in the absence of CAFs. PCA analysis of the metrics generated from model runs without CAFs shows that the data reduces to a single dimension (*Figure 2—figure supplement 1f* and Appendix 3 – table 1 in *Supplementary file 1*), with remarkably similar behaviour in both organotypic and spheroid data. PCA combining runs with and without CAFs confirmed that fibroblasts boost invasion (*Figure 2—figure supplement 1*).

## Matrix proteolysis drives strand widening but not the extent of invasion

Having established that our model could generate diverse types of invasion, we undertook a more systematic analysis of parameter space to determine the contribution of specific parameters to both the extent and pattern of invasion. *Figure 3a* shows the PCA plots overlaid with shading for the input variable of cancer cell proteolysis, with high levels of proteolysis, trending along the vector for number of neighbours in both organotypics and spheroids. Somewhat contrary to expectation, we found that increasing cancer cell proteolysis led to only modestly elevated invasion scores in organotypic contexts. Moreover, the maximum invasive depth did not correlate with matrix proteolysis (*Figure 3b*). Instead the width of the strands (neighbour numbers) increased as a function of proteolysis, especially in organotypic assays. In simulations with low proteolysis, the model predicted thin strands (low neighbour and low tapering scores). To measure the effect of proteolysis on the shape of the invading front of cell clusters, we ran simulations initiated with a cluster of cells and a uniform directional cue, either without the complicating factor of pre-existing tracks or a simple single permissive track. *Figure 3—figure supplement 1c* shows that increasing proteolysis leads to reduced curvature and a 'pushing' front in the absence of a track. When a track was present, it was favoured for invasion and interfered with the generation of a pushing front most strikingly at intermediate levels of proteolysis. Inspired by previous studies (*Ahmadzadeh et al., 2017*; *Park et al., 2020*; *Provenzano et al., 2006*), we additionally considered the cases if the ECM had multiple tracks either oriented parallel to the direction of invasion – analogous to aligned matrix fibres – or had isotropic texture distributed as a chessboard – analogous to non-aligned matrix fibres. As might be expected, ECM fibres parallel to the direction of the invasive cue favoured invasion, but isotropic texture hindered invasion (*Figure 3—figure supplement 1d*).

Analysis of spheroid contexts yielded a different picture, with reduced maximum invasion depth with increasing proteolysis values. Notably, the very highest matrix degradation value yielded significantly lower maximum invasion depth than the intermediate and lowest level. There was less difference in the overall invasion score as increasing proteolysis was linked to slightly wider strands, which counter-balanced the reduction in maximum invasion depth (*Figure 3a and b*). Both organotypic assays and spheroids without CAFs exhibited low levels of invasion (*Figure 3—figure supplement 1a* and b). Comparative plots of the metrics in simulations with and without CAFs confirm this (note the red colour) and indicate that fibroblasts favour narrower strands (note the blue colour in the neighbour and tapering rows). Overall, cancer cell proteolysis is primarily predicted to regulate strand width in both organotypic and spheroid contexts. The relationship between matrix proteolysis and strand

**Table 1.** Key CC3D parameter values.

| CC3D parameter | CC3D parameter value | Real world value | Comments |
|---|---|---|---|
| Monte Carlo timestep (MCS) | 1 | 30 s | |
| Voxel | 1 | 2 microns | |
| VCAF target volume | 800 voxels | 6400 microns$^3$ | 6500 microns$^3$ experimental measure |
| VCAF target surface | 700 voxels | 2800 microns$^2$ | 4900 microns$^2$ experimental measure |
| SCC initial target volume | 400 voxels | 3200 microns$^3$ | |
| SCC dividing volume | 800 voxels | 6400 microns$^3$ | |
| Median SCC volume in wild-type conditions | 550 voxels | 4400 microns$^3$ | 4500 microns$^3$ experimental measure |
| SCC surface area | 324 voxels (median) | 1296 microns$^2$ (median) | Sphere assumed for surface area 1700 microns$^2$ experimental measure |
| Mean time to mitosis | 8640 MCS | 3 days | |
| SCC-ECM adhesion | 10 (contact energy) | 45.53 (experimental measure) | Adhesions are normalised to SCC-ECM adhesion. They are inverted and multiplied by 10 to give contact energies |
| SCC-SCC adhesion | 21 (contact energy) | 21.8 | |
| SCC-CAF adhesion | 35 | 25.2 | SCC-CAF adhesion was marginally reduced below experimental measure in model (contact energy would be 18 from experiment) |
| CAF-CAF adhesion | 45 | 9.3 | |
| CAF-ECM adhesion | 15 | 29.6 | |
| SCC-ECM adhesion for zero-density ECM | 40 | 11.4 | |
| CAF-CAF repulsion range | 20 voxels | 40 microns | Approximately 1.5 CAF widths |
| SCC taxis energy | 13 | median speed 0.2 microns/min | 0.2 microns/min experimental measure |
| CAF taxis minimum energy | 3.5 | 0.06 micron/min | |
| CAF taxis maximum energy | 21 | 0.29 microns/min | |
| CAF median speed | 10 | 0.1 microns/min | 0.1 microns/min experimental measure |
| CAF taxis stimulation range | 30 voxels | | Approximately 2.5 CAF widths |
| CAF ECM pushing rate | 0.0140 | Corresponds to a reduced CAF speed of 0.07 microns/min through ECM | Speed due to pushing is three times faster than speed due to degradation |
| CAF ECM degradation rate | 0.0012 | Corresponds to a reduced CAF speed of 0.018 microns/min through ECM | Degradation and pushing effects on speed are sub-linear. The effective median speed is between 0.07 and 0.088 microns/min |
| SCC pushing rate | 0 | | |
| SCC degradation rate | 0.0001 | Corresponds to a reduced SCC speed of 0.009 microns/min through ECM | Effect of degradation is half for SCCs compared to CAFs. Effects are normalised to kinesis levels |

CAF, cancer-associated fibroblast; ECM, extracellular matrix; SCC, squamous cell carcinoma; VCAF, vulval CAF.

width was maintained even if cancer cell proliferation was reduced (*Figure 3—figure supplement 1e*), although the cell neighbour values were lower when proliferation was halved.

We tested the predictions that cancer cell matrix protease function was linked to width of invasion strands by generating A431 cancer cells that either over-expressed *MMP14*, the major collagen protease, or had it deleted via Crispr/Cas9 editing methods (*Figure 3—figure supplement 1f*). In line with expectation, MMP14 over-expression increased the proteolysis of collagen, while MMP14

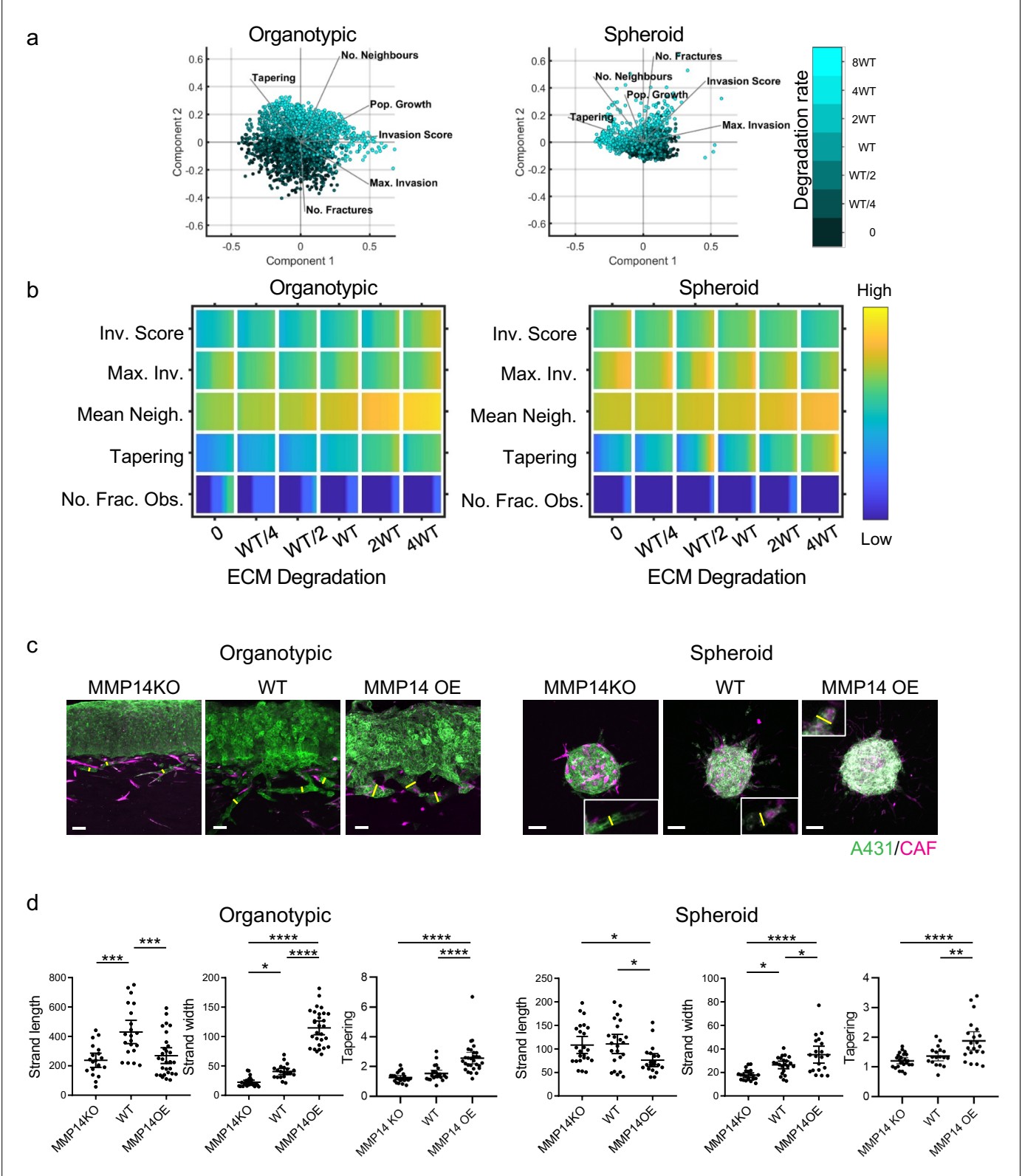

**Figure 3.** Matrix proteolysis determines strand width but not the distance invaded. (**a**) Principal component analysis plots show the metrics derived from over 2000 simulations in the presence of fibroblasts covering variation in cancer cell proteolysis with values indicated by the intensity of cyan, cancer cell–matrix adhesion (not colour coded), and cancer cell–cancer cell adhesion (not colour coded). (**b**) Heatmaps show how varying the cancer cell proteolysis value (x axis) impacts on different metrics when fibroblasts are included in all simulations. WT indicates the 'wild-type' value based on

*Figure 3 continued on next page*

*Figure 3 continued*

experimental parameterisation using A431 cancer cells. Yellow indicates a high value, dark blue a low value. (**c**) Images show the effect of modulating matrix proteolysis via either MMP14 Crispr KO or MMP14 over-expression in cancer cells (green) both organotypic and spheroid assays including fibroblasts (magenta). Scale bar = 100 μm. (**d**) Quantification of three biological replicates of the experiment shown in panel (c) with strand length, strand width, and tapering shown – 1 unit is equivalent to 0.52 μm. Error bars indicate 95% confidence intervals, one dot represents one strand. ECM, extracellular matrix.

The online version of this article includes the following source data and figure supplement(s) for figure 3:

**Source data 1.** Quantification of invading strand length, width, and tapering in A431 cells with/without MMP14 manipulation.

**Figure supplement 1.** Matrix proteolysis determines strand width but not the distance invaded.

**Figure supplement 1—source data 1.** Quantification of invading strand width in A431 WT and MMP14 OE cells pretreated with mitomycin C.

**Figure supplement 1—source data 2.** Uncropped western blot images of WT, MMP14 KO, MMP14 OE, CTNNA1 KO, MMP14 KO/CTNNA1 KO, and MMP14 OE/CTNNA1 KO A431 lysates stained for MMP14, alpha-catenin, vimentin, fibronectin, or β-actin.

deletion reduced proteolysis (as assessed by DQ collagen fluorescence) (*Figure 3—figure supplement 1g*). *Figure 3c and d* shows that experimentation confirmed the major predictions of our model. In particular, the maximum invasion depth in the organotypic context did not simply increase with MMP14 levels, with strand lengths similar between *MMP14* KO and over-expressing cells. In contrast, the strand width was notably affected by MMP14 levels in both organotypic and spheroid assays (yellow lines in *Figure 3c* indicate strand width), with KO cells generating thin strands and over-expressing cells generating thick strands. The positive relationship between ECM proteolysis and strand width was particularly strong in organotypic contexts (*Figure 3c and d*). Of note, matrix proteolysis promoted wide strands even if cancer cell proliferation was prevented by pre-treatment with mitomycin C (*Figure 3—figure supplement 1h*). These results are highly concordant with the model predictions and confirm that MMP14 is a major determinant of the mode of collective cancer cell invasion but plays little role in determining the maximum distance invaded.

## Cancer cell-cell adhesion promotes wide invasive strands

We turned our attention to investigate how cancer cell adhesion to either other cancer cells or the matrix influenced the mode of collective invasion. *Figure 4a* shows PCA plots of invasion characteristics with the strength of cancer cell–matrix adhesion overlaid in green shading. There was no consistent association between cancer cell-matrix adhesion and invasive pattern in the organotypic context, with high adhesion values distributed across the PCA plot. The relationship between cell-matrix adhesion and invasion score was relatively flat, with only very high cell-matrix adhesion values boosting invasion. This prediction is supported by the lack of effect of *ITGB1* deletion on cancer invasion in the experimental organotypic model (*Figure 4c and d* – *Figure 4—figure supplement 1b* and c confirm that ITGB1 KO cells are defective in collagen I and Matrigel adhesion). In the spheroid context, there was a somewhat stronger association between matrix adhesion and invasion. Minimal invasion was observed in the absence of fibroblasts (*Figure 4—figure supplement 1a*). Intriguingly, the strongest correlation was with the tapering metric that reflects whether strands have a uniform breadth or taper as they invade deeper (*Figure 4b* – row 4). Experiments using *ITGB1* KO A431 cells provided support for this prediction. To rule out a compensatory role for ITGB3 in ITGB1 KO cells, we combined targeting of both ITGB1 and ITGB3. *Figure 4—figure supplement 1d–f* shows that these cells were still able to invade. Greater tapering observed in *ITGB1* KO spheroids (*Figure 4c and d*). Interestingly, and in line with model predictions, this was not observed in organotypic assays (*Figure 4b–d*).

Next, we explored the relationship between cancer cell–cancer cell adhesion and invasion when fibroblasts were present (*Figure 5a*). These analyses yielded several predictions that caught our attention. First, reducing cancer cell–cancer cell adhesion reduced the total invasion score in organotypic assays across relatively large ranges of parameter space (*Figure 5a and b* – note the association of increasing magenta intensity and invasion score vectors in the PCA plot). This is counter to the widely held view that EMT and increased single cell characteristics promote invasion. Specifically, in organotypic contexts, lower cancer cell–cancer cell adhesion resulted in shorter invasive strands that thinned rapidly as they invaded (this is reflected in the Max. Invasion, Mean Neighbour, and Tapering rows in *Figure 5b*). Once again, little invasion was observed in the absence of fibroblasts (*Figure 5—figure supplement 1a*). *Figure 5—figure supplement 1c* explicitly plots the change in strand width as a

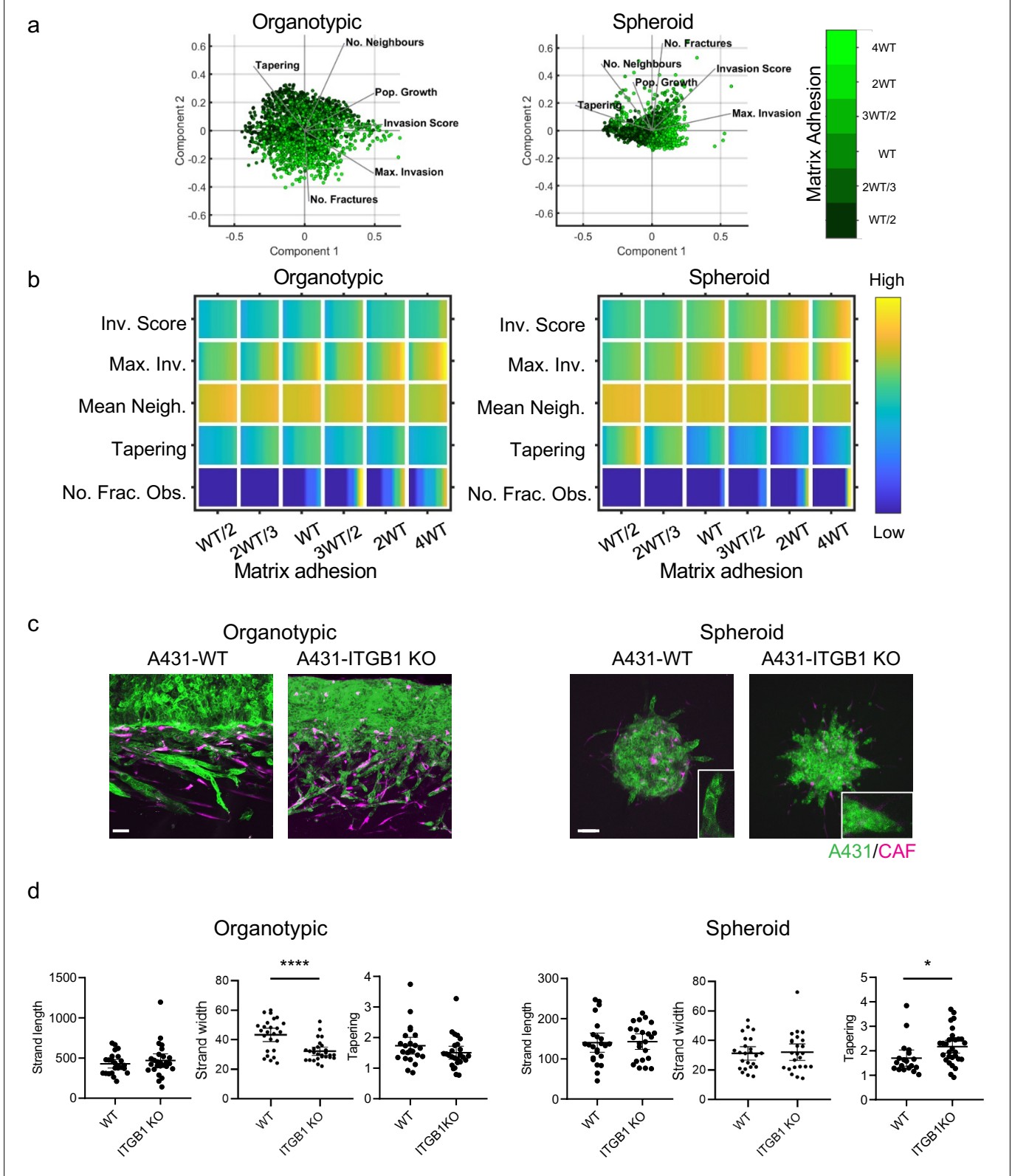

**Figure 4.** Cancer cell-matrix adhesion modulates the tapering of strands. (**a**) Principal component analysis plots show the metrics derived from over 2000 simulations in the presence of fibroblasts covering variation in cancer cell–matrix adhesion with values indicated by the intensity of green, cancer cell proteolysis (not colour coded), and cancer cell–cancer cell adhesion (not colour coded). (**b**) Heatmaps show how varying the cancer cell–matrix adhesion value (x axis) impacts on different metrics when fibroblasts are included in all simulations. WT indicates the 'wild-type' value based on

*Figure 4 continued on next page*

Figure 4 continued

experimental parameterisation using A431 cancer cells. Yellow indicates a high value, dark blue a low value. (**c**) Images show the effect of modulating matrix adhesion via Crispr KO of ITGB1 in cancer cells (green) in both organotypic and spheroid assays including fibroblasts (magenta). Scale bar = 100 μm. (**d**) Quantification of three biological replicates of the experiment shown in panel (c) with strand length, strand width, and tapering metric shown – 1 unit is equivalent to 0.52 μm. Unpaired t-test was performed. 95% confidence intervals are shown, one dot represents one strand.

The online version of this article includes the following source data and figure supplement(s) for figure 4:

**Source data 1.** Quantification of invading strand length, width, and tapering in A431 cells with/without ITGB1 manipulation.

**Figure supplement 1.** Cancer cell–matrix adhesion modulates the tapering of strands.

**Figure supplement 1—source data 1.** Quantification of ECM adhesion in A431 WT and ITGB1 KO cells.

**Figure supplement 1—source data 2.** Uncropped western blot images of WT, ITGB1 KO, ITGB1 KO/control KD, and ITGB1 KO/ITGB3 KD A431 lysates stained for integrin β1, integrin β3, or β-actin.

function of depth for varying cancer cell–cancer cell adhesion. The simpler context of cell invasion into a thin permissive gap further supported the prediction that cancer cell–cancer cell adhesion is linked to wider invading strands (*Figure 5—figure supplement 1d*). The situation in spheroid assays was more subtle, with increases in invasion only predicted at very high values ≥2 WT (*Figure 5b*). Of note, the Neighbour and Tapering metrics did not vary much depending on cancer cell–cancer cell adhesion. To test these predictions, we generated A431 cells defective in cell–cell adhesion as a result of Crispr-mediated deletion of α-catenin/*CTNNA1* (*Figure 3—figure supplement 1f*). Strikingly, and, in line with the model predictions, these cells lacking adherens junctions were significantly less invasive, both in terms of strand length and strand width, in organotypic assays (*Figure 5d*). In spheroid assays, loss of α-catenin did not affect strand length and had only a modest effect on strand width (~20% reduction compared to a 60% reduction in width in organotypic assays).

## The pro-invasive role of cell-cell junctions depends on a uniform directional cue and supra-cellular coordination of the actomyosin cytoskeleton

The data described above establish an intriguing context-dependent role for cell-cell junctions in collective invasion – with a positive relationship between cell-cell adhesion and invasion in organotypic contexts but not in spheroid contexts. To rule out that the location of CAFs drive invasive pattern, we mixed CAFs with SCCs in organotypic simulations (*Figure 5—figure supplement 1e*). The results were highly consistent with the results of CAFs mixed in with ECM, with only minor differences in the number of fractured objections and tapering. These analyses rule out CAF location as a dominant driver of invasive pattern. One key difference between these two contexts is that cancer cells in the organotypic context are subject to a uniform gradient of chemotactic cues, whereas in the spheroid context, the cancer cells are subject to a radial chemotactic cue. We used our model to test if switching to a uniform chemotactic gradient in the spheroid context would generate a positive relationship between cell-cell adhesion and invasion. *Figure 5e* quantifies track invasion score in simulations of spheroids with either uniform or radial chemotactic cues. These analyses indicate that cancer cell junctions are favourable for invasion when cells are subject to a uniform directional cue. The importance of junctions only when there is a uniform directional cue suggests that it may not be cell-cell adhesion per se that is important but some linkage between cell-cell adhesions and coordination of collective invasion. Consistent with this idea, cadherin-mediated coordination of actin and myosin dynamics is important for effective collective migration of neural crest cells during cranio-facial development and for border cell migration in the *Drosophila* egg chamber (*Geisbrecht and Montell, 2002*; *Shellard et al., 2018*). We hypothesised that a similar mechanism might also underlie the context-dependent importance of adherens junctions in cancer cell invasion.

Previous work revealed that collectively invading cancer cells have a supra-cellular actomyosin network that enables the coordinated migration of cell groups. *Figure 6a* confirms control A431 cells exhibit supra-cellular organisation of their actomyosin network (*Hidalgo-Carcedo et al., 2011*). Furthermore, knockout of *CTNNA1* disrupts the formation of a supra-cellular actomyosin network (*Figure 6a* – quantification shown in *Figure 6—figure supplement 1a*). As expected, CTNNB1 failed to localise to cell-cell contacts in CTNNA1 KO A431 cells (*Figure 6a*). To experimentally disrupt the supra-cellular actomyosin network while retaining cell-cell junctions, we utilised two experimental

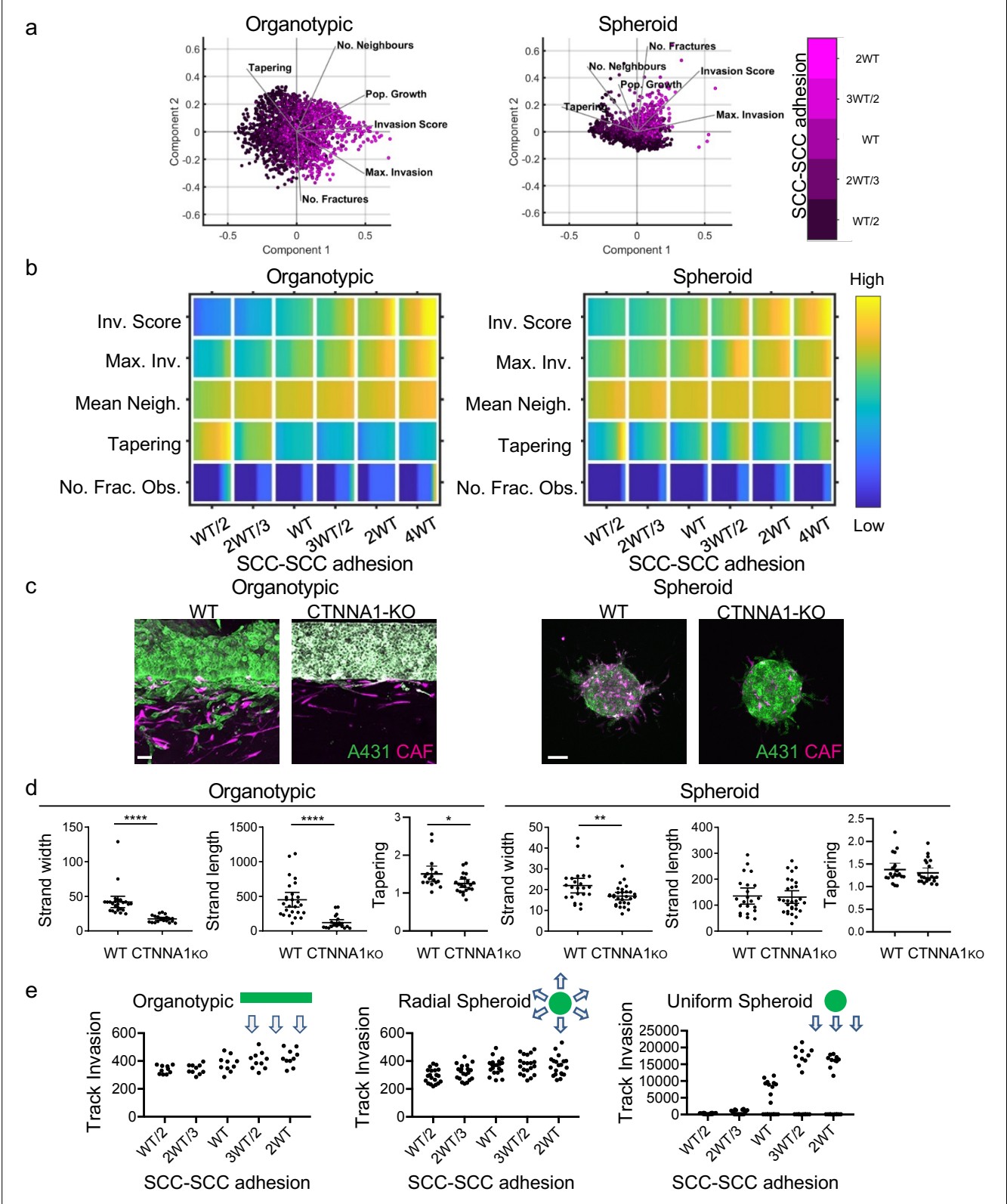

**Figure 5.** Cancer cell-cancer cell adhesion is required for efficient invasion in response to uniform directional cues. (**a**) Principal component analysis plots show the metrics derived from over 2000 simulations in the presence of fibroblasts covering variation in cancer cell–cancer cell adhesion with values indicated by the intensity of magenta, cancer cell proteolysis (not colour coded), and cancer cell–matrix adhesion (not colour coded). (**b**) Heatmaps show how varying the cancer cell–cancer cell adhesion value (x axis) impacts on different metrics when fibroblasts are included in all

*Figure 5 continued on next page*

*Figure 5 continued*

simulations. WT indicates the 'wild-type' value based on experimental parameterisation using A431 cancer cells. Yellow indicates a high value, dark blue a low value. (**c**) Images show the effect of modulating cancer cell-cell adhesion via Crispr KO of CTNNA1 in cancer cells (green) in both organotypic and spheroid assays including fibroblasts (magenta). Scale bar = 100 μm. (**d**) Quantification of three biological replicates of the experiment shown in panel (c) with strand length, strand width, and tapering shown – 1 unit is equivalent to 0.52 μm. Unpaired t-test was performed. Error bars indicate 95% confidence intervals, one dot represents one strand. (**e**) Plots show the track invasion score with varying cancer cell–cancer cell adhesion in simulations lacking fibroblasts but with a single permissive track favouring invasion. Cartoons indicate the initial set up of cell positions and the directional cue in the simulation.

The online version of this article includes the following source data and figure supplement(s) for figure 5:

**Source data 1.** Quantification of invading strand length, width, and tapering in A431 cells with/without CTNNA1 manipulation.

**Figure supplement 1.** Cancer cell–cancer cell adhesion is required for efficient invasion in response to uniform directional cues.

tools, ROCK:ER (a fusion of the ROCK2 kinase domain to the regulatory domain of the oestrogen receptor) and ROCK kinase inhibition. In the presence of 4OHT, ROCK:ER boosts actomyosin contractility throughout the cytoplasm including at cell-cell interfaces (*Croft et al., 2004*), and the latter reduces the activity of the supra-cellular actomyosin belt. *Figure 6b and c* shows that these manipulations have the desired effect on active actomyosin, as determined by pS19-MLC staining (*Figure 6—figure supplement 1b* and c confirms these observations with staining for MYH9). We next tested the effect of these perturbations on A431 *MMP14* over-expressing cells that generate wide invasive strands. *Figure 6d and e* shows that both manipulations reduce the width of invading strands, demonstrating that disrupting actomyosin coordination mechanisms phenocopy loss of adherens junctions with respect to the width of invading strands (note: for this experiment we used ECM pre-conditioned by fibroblasts in the absence of drug and then added cancer cells in the presence of the indicated perturbations). Furthermore, the data support a model in which adherens junctions influence invasive pattern by enabling supra-cellular coordination of actomyosin, and not simply determining whether cancer cells are able to maintain contact with one another. Consistent with this view, we observed supra-cellular organisation of actomyosin and the retention of adherens in all but the thinnest invading strands in human SCC.

## Protease-driven strand widening requires cell-cell junctions

The analyses above investigate the relationship between individual cancer cell parameters and invasion; we additionally explored how combinations of parameter variations influenced invasive pattern and extent. The data described above argue that, by virtue of their role in coordinating supra-cellular actomyosin, cell-cell junctions would be required for high levels of proteolysis to generate wide invasive tracks. We, therefore, explored the interplay between cancer cell–cancer cell adhesion and proteolysis in determining SCC invasion using both modelling and experimental strategies. Potts modelling predicted that the high neighbour number observed when matrix proteolysis is high would depend upon cell-cell junctions in organotypic assays (note the higher values in the top right regions on the plots in *Figure 7ai*). Interestingly, this cooperative interaction between proteolysis and cell-cell adhesion was not predicted to influence the extent of maximum invasion, which was dominated by cell-cell adhesion alone (*Figure 7aii*). These predictions were supported by experimentation: deletion of *CTTNA1* prevented the formation of wide invasive strands by *MMP14* over-expressing A431 cells in the organotypic invasion assays (*Figure 7c*), with more subtle effects observed in the spheroid assays (*Figure 7a, d, and e*). *Figure 3—figure supplement 1a* and b indicates that CAFs favour narrower invasive strands; therefore, to more fully explore how ECM proteolysis and cell-cell adhesion co-ordinately determine the geometry of collective invasion, we revisited simulations, without CAFs, designed to monitor the curvature of the invading cell cluster (*Figure 3—figure supplement 1d*). *Figure 7—figure supplement 1a* and b shows that if both ECM proteolysis and cell-cell adhesion are high then a broad, virtually flat, invasive front is generated. Reducing either proteolysis or cell-cell adhesion leads to increased curvature. Together, these analyses establish that a broad 'pushing' front of invasion requires both high proteolysis and high cancer cell–cancer cell adhesion.

## Strand widening is coupled to cancer cell growth

While the focus of our analysis has been the pattern of invasion, the widening of tracks might also represent a mechanism for generating additional space for cell growth in confined environments. As

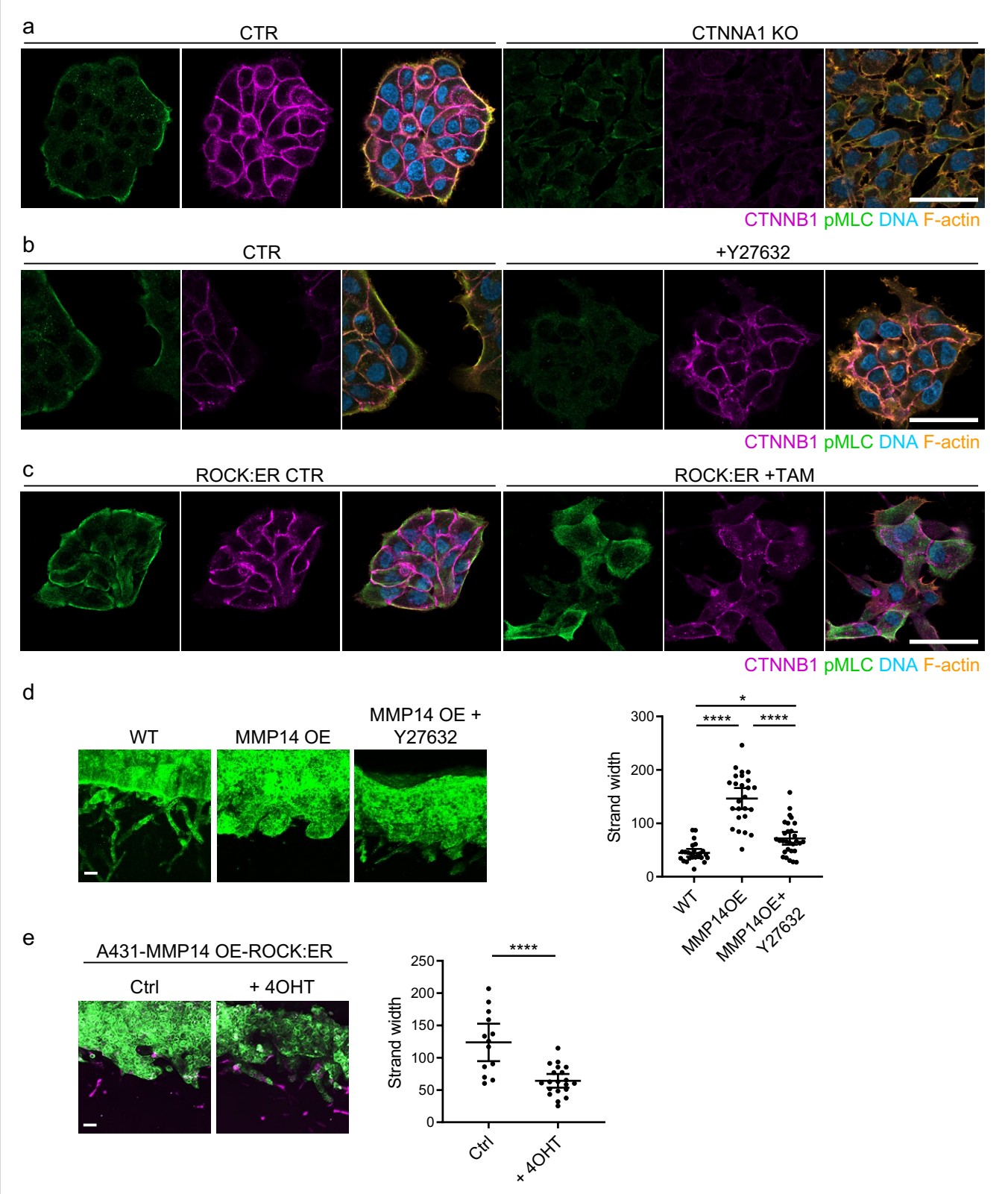

**Figure 6.** Supra-cellular coordination of actomyosin organisation by cell–cell junctions enables wide invading strands. (**a**) Images show the β-catenin (magenta), F-actin (orange), DNA (blue), and active myosin (pS19-MLC - green) networks in control A431 and CTNNA1 KO A431 cells.( **b**) Images β-catenin (magenta), F-actin (orange), DNA (blue), and active myosin (pS19-MLC - green) networks in control A431- and 10-μM Y27632-treated cells. Scale bar = 20 μm. (**c**) Images show β-catenin (magenta), F-actin (orange), DNA (blue), and active myosin (pS19-MLC - green) networks in control A431

*Figure 6 continued on next page*

*Figure 6 continued*

ROCK:ER- and 4-OHT-treated cells. Scale bar = 20 μm. (**d**) Images show organotypic killing assays using control or MMP14 over-expressing A431 cells in the presence or absence of 10 μM Y27632. Scale bar = 100 μm. Plot shows the quantification of strand width from three biological replicates – 1 unit is equivalent to 0.52 μm. One-way ANOVA with post-hoc multiple comparisons was performed. Error bars indicate 95% confidence intervals, one dot represents one strand. (**e**) Images show organotypic invasion assays using MMP14 over-expressing A431 cells additionally engineered to contain ROCK:ER in the presence or absence of 4-OHT. Scale bar = 100 μm. Plot shows the quantification of strand width from three biological replicates. Unpaired t-test was performed. Error bars indicate 95% confidence intervals, one dot represents one strand.

The online version of this article includes the following source data and figure supplement(s) for figure 6:

**Source data 1.** Quantification of invading strand width in A431 cells with/without manipulation of actomyosin contractility.

**Figure supplement 1.** Supra-cellular coordination of actomyosin organisation by cell–cell junctions enables wide invading strands.

**Figure supplement 1—source data 1.** Quantification of pMLC intensity in A431 WT, CTNNA1 KO, and cells with actomyosin manipulation.

proliferation is a feature of our model, we additionally investigated whether cancer cell growth might be impacted as a result of change in cancer cell–cancer cell adhesion and proteolysis. Interestingly, the vectors reflecting cell growth and neighbour number in the PCA analysis were closely aligned (*Figure 2c and d*). EdU staining revealed that proliferating cells were observed throughout spheroids (*Figure 8—figure supplement 1a and b*). Moreover, there was a positive association between the proportion of EdU positive cells in invading strands and the strand width. The modelling indicated that the linkage between strand widening and growth was particularly pronounced in the spheroid simulations lacking fibroblasts (*Figure 2—figure supplement 1f*). Given that we have established matrix proteolysis and cancer cell–cancer cell adhesion as the major determinants of neighbour number and strand width, we therefore investigated the relationship between these parameters and cell growth. Neither was predicted to have a strong effect on cell growth in organotypic assays, either in the presence or absence of CAFs (*Figure 8a*). In contrast, a strong positive relationship between proteolysis and growth was predicted in the context of spheroids lacking CAFs (*Figure 8a*). Cancer cell–cancer cell adhesions were also predicted to make a positive contribution to growth, albeit smaller than the effect of proteolysis (*Figure 8a*). We proceeded to test these predictions experimentally. Manipulation of *MMP14* and *CTTNA1* had minimal effect on cell growth in unconfined two-dimensional (2D) culture conditions (*Figure 8—figure supplement 1c*). *Figure 8b and c* confirms that both proteolysis and cancer cell–cancer cell adhesion are required for effective cell growth in 3D collagen matrices. Moreover, the positive effect of boosting proteolysis required cell-cell adhesions (*Figure 8b and c* compares MMP14 OE with αCATKO MMP14 OE). Ectopic activation of ROCK2, which disrupts cytoskeletal cohesion in cell clusters, also reduced growth in 3D collagen (*Figure 8—figure supplement 1d* and e). Together, these data suggested that the supra-cellular actomyosin network, invasive strand width, and cancer cell growth might be linked.

The linkage between strand widening and growth might be due to the ability of cells to generate space when surrounded by ECM. This could be the result of proteolysis, which would explain the effect of MMP14 manipulation, but it is less clear why this might require adherens junctions. We hypothesised that cell-cell junctions and the supra-cellular coordination of the actomyosin network might enable cancer cells to physically remodel the ECM (*Figure 8d–f*). Similar to previous work with single cells *Wyckoff et al., 2006*, we observed that clusters of control cancer cells displaced the ECM. This was observed directly in time-lapse movies and as the formation and compaction of ECM fibres at the cancer cell-ECM interface (note arrows in *Figure 8e and f*). These analyses also revealed highly dynamic membrane blebs and filopodia at the cancer cell-ECM interface. ECM compaction was absent when CTNNA1 KO cells were used. MMP over-expression reduced ECM compaction and led to the formation of gaps in the ECM adjacent to the cancer cells with reduced numbers of membrane blebs (*Figure 8f*). These analyses provide a direct demonstration of the 'pushing' term used in the computational modelling and are consistent with a role for the supra-cellular actomyosin cable in generating the pushing force.

## Protease-driven tumour growth and lymph node metastasis require cell-cell junctions

Finally, we sought to test whether key findings of our integrated in silico and in vitro analysis also applied in an in vivo context with a heterogeneous environment including a greater diversity of

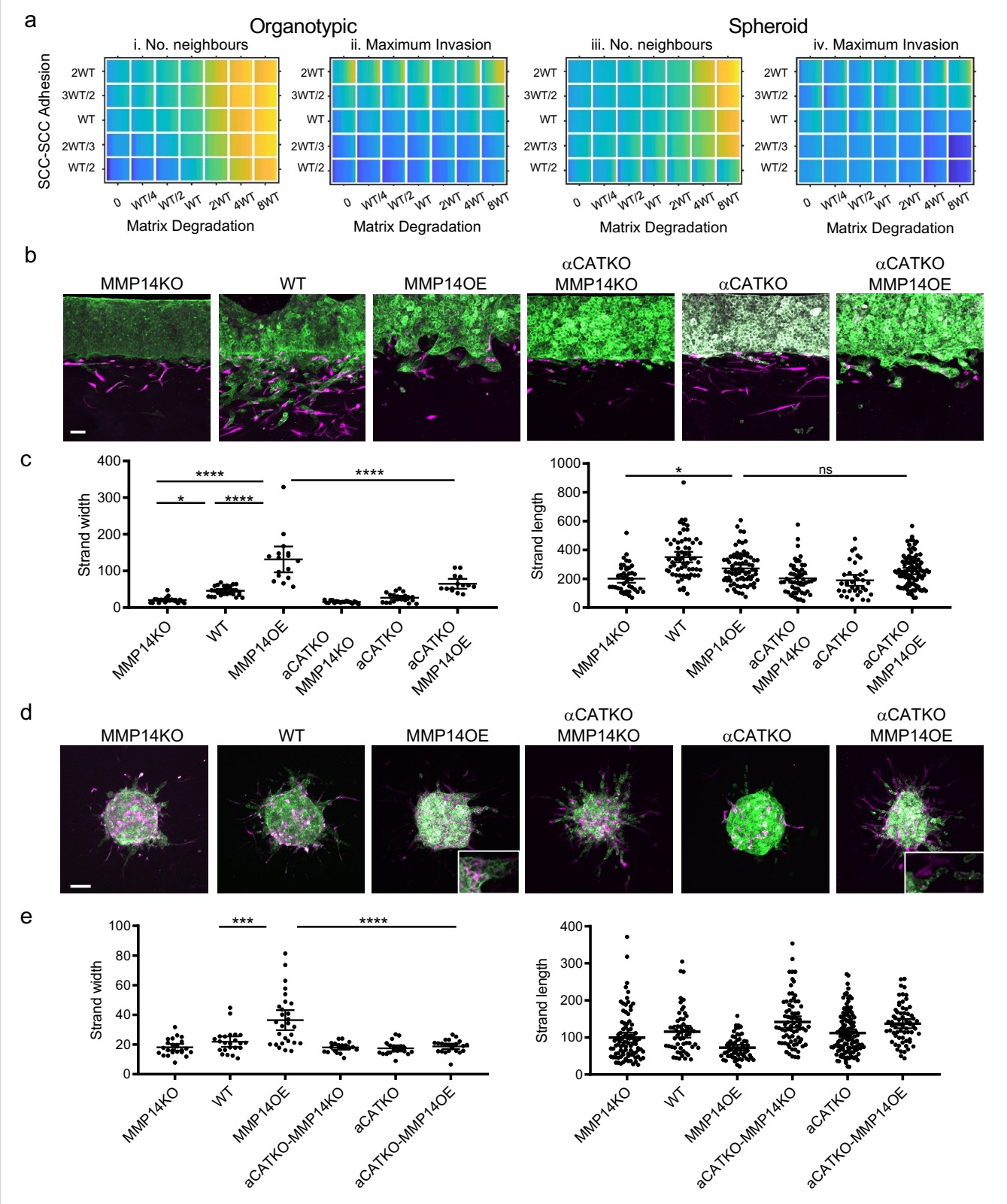

**Figure 7.** Proteolysis-driven strand widening requires adherens junctions. (**a**) Heatmaps show how varying the matrix proteolysis (x-axis) and cancer cell–cancer cell adhesion value (y axis) impacts on different metrics when fibroblasts are included in all simulations. WT indicates the 'wild-type' value based on experimental parameterisation using A431 cancer cells. Yellow indicates a high value, dark blue a low value. (**b**) Images show the effect of combinatorial modulation of matrix proteolysis and cancer cell-cell adhesion via Crispr KO of CTNNA1 and/or MMP14 and/or MMP14 over-expression

*Figure 7 continued on next page*

*Figure 7 continued*

in cancer cells (green) in both organotypic assays including fibroblasts (magenta). Scale bar = 100 µm. (**c**) Quantification of three biological replicates of the experiment shown in panel (b) with strand length and strand width shown – 1 unit is equivalent to 0.52 µm. One-way ANOVA with post-hoc multiple comparisons was performed. Error bars indicate 95% confidence interval, one dot represents one strand. (**d**) Images show the effect of combinatorial modulation of matrix proteolysis and cancer cell-cell adhesion via Crispr KO of CTNNA1 and/or MMP14 and/or MMP14 over-expression in cancer cells (green) in both spheroid assays including fibroblasts (magenta). (**e**) Quantification of three biological replicates of the experiment shown in panel (d) with strand length and strand width shown. Scale bar = 100 µm. One-way ANOVA with post-hoc multiple comparisons was performed. Error bars indicate 95% confidence interval, one dot represents one strand.

The online version of this article includes the following source data and figure supplement(s) for figure 7:

**Source data 1.** Quantification of invading strand width and length in A431 cells with/without manipulation of MMP14 and/or CTNNA1.

**Figure supplement 1.** Proteolysis-driven strand widening requires adherens junctions.

stromal cell types not included in our model. A431 cells engineered to have different levels of *MMP14* and *CTTNA* levels were injected into the dermal space within the ears of mice. This anatomical location was chosen because the dermis represents the first tissue that SCC invades into, and cells can spread from the dermis to local lymph nodes, which reflects the clinical spread of the disease. This environment is spatially confined with some fibroblasts in addition to thin layers of fat, cartilage, and muscle. It was not possible to include all these additional factors with appropriately controlled parameterisation. Therefore, we concentrated on validating the relationship between matrix proteolysis, cancer cell-cell adhesion, and invasive spread in vivo. In addition, if stromal support, such as that provided by fibroblasts, is limited then the mechanisms that promote wide invasive strands also favour growth. To test these ideas, we injected A431 cells with combinations of MMP14 and α-catenin manipulations into the intradermal space of mouse ears. This environment is spatially restrictive with lymphatic drainage to local lymph nodes. Of note, MMP over-expressing cells generated tumours with particularly wide, bulging, margins (*Figure 9a*). Strikingly, there was a strong correlation between the levels of MMP14 and tumour growth (*Figure 9b*). Histological analysis revealed clusters of SCC cells in the ear distant from the main tumour. In MMP14 over-expressing tumours, these clusters were larger, rounder (as judged by aspect ratio), and further from the tumour (*Figure 9—figure supplement 1a* and b). Metastatic spread to lymph nodes also correlated with MMP14 levels, which are in line with previous reports (*Bartolomé et al., 2009*; *Devy et al., 2009*; *Wang et al., 2021*). Notably, and in contrast to the prevailing dogma, reducing cancer cell–cancer cell adhesion did not lead to a more aggressive tumour phenotype but reduced both tumour growth, and very few mice were observed to have lymph node metastases (*Figure 9c*). This could be partly compensated by over-expression of MMP14, suggesting that a defect in 'space' generation might underpin the defect in the *CTNNA1* KO cells. However, the growth and lymph node metastasis of *MMP14* o.e./*CTNNA1* KO cells were reduced compared to the *MMP14* o.e. cells (*Figure 9c*), indicating that the tumour promoting effect of elevated MMP14 levels depends on cell-cell adhesion. Together, these analyses demonstrate that MMP14-driven matrix proteolysis promotes invasion in wide collective units and tumour growth in spatially confined contexts. Furthermore, the widening of invasive units, tumour growth, and lymph node metastases depends upon adherens junction-mediated supra-cellular coordination of the actomyosin network.

## Discussion

The combined computational and experimental analysis of collective cancer cell invasion presented here raises several findings that warrant further consideration. Although, matrix proteolysis was broadly associated with higher levels of invasion (*Castro-Castro et al., 2016*; *Egeblad and Werb, 2002*), it was not a simple linear relationship (*Figure 8—figure supplement 1d*). Most notably, high proteolysis reduces the maximal extent of invasion but increases the strand width in both the model and experiments. The ability of cells with high levels of proteolysis to generate space means that there is less pressure to constrict cells into longer thinner strands. The importance of space limitation for effective invasion is underscored by the reduced invasion observed when spheroids have a 'choice' between invasion and spreading over an unimpeded matrix layer. High proteolysis essentially reduces the space limitation. This is also linked with high levels of proliferative capacity in 3D environments. Our analysis demonstrated that this growth effect was clearly observed in vivo. *MMP14*

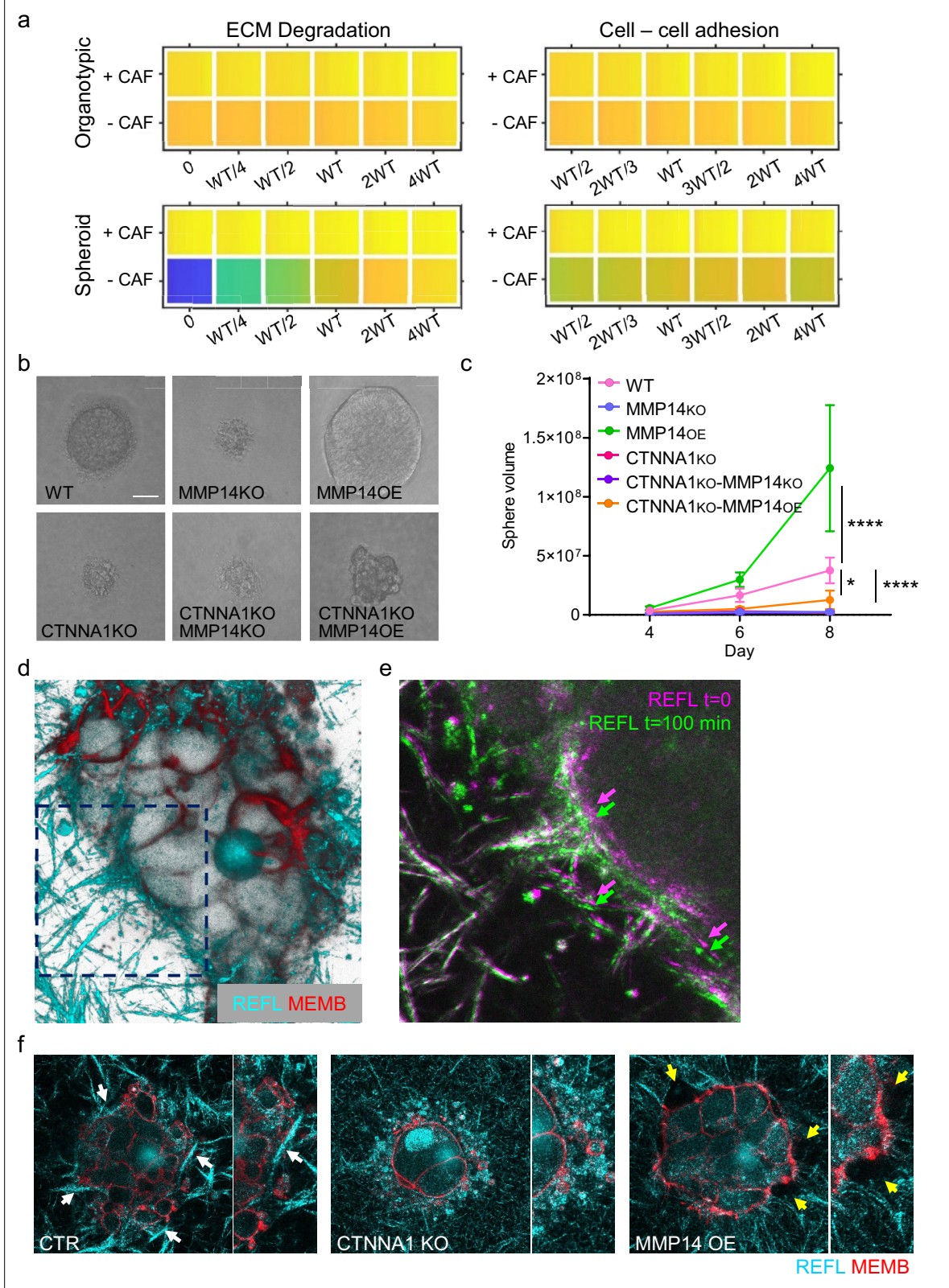

**Figure 8.** Strand widening is linked to tumour growth and metastasis. (**a**) Heatmaps show how varying the matrix proteolysis (left) or cancer cell–cancer cell adhesion value (right) impacts on predicted cell growth in the presence or absence of fibroblasts. WT indicates the 'wild-type' value based on experimental parameterisation using A431 cancer cells. Yellow indicates a high value, dark blue a low value. (**b**) Phase contrast images show the growth of cancer cell colonies with the indicated manipulations of MMP14 and CTNNA1 after 8 days surrounded by matrix. Scale bar = 50 µm. (**c**) Plot

*Figure 8 continued on next page*

*Figure 8 continued*

shows quantification of the growth assay shown in (b). Two-way ANOVA with post-hoc multiple comparisons was performed. Error bars indicate 95% confidence intervals. Data from three biological replicates. (**d**) Fluorescent image shows reflectance of collagen fibre (cyan) and cell membrane of A431 WT cells in three-dimensional (3D) culture. (**e**) Fluorescent image shows reflectance of collagen fibres around A431 WT cells in 3D culture at two time points. t=0 min: magenta, t=100 min: green. (**f**) Fluorescent images show reflectance of collagen fibres (cyan) and cell membrane of A431 WT, CRNNA1 KO, or MMP14 over expressing cells (red) in 3D culture. White arrows highlight the formation and motion of collagen bundles adjacent to the cell clusters, yellow arrows highlight gaps.

The online version of this article includes the following source data and figure supplement(s) for figure 8:

**Source data 1.** Quantification of cancer cell proliferation in 3D culture.

**Figure supplement 1.** Strand widening is linked to tumour growth and metastasis.

**Figure supplement 1—source data 1.** Quantification of proliferation of WT, MMP14, CTNNA1, and/or ROCKER manipulated A431 in 2D and 3D culture.

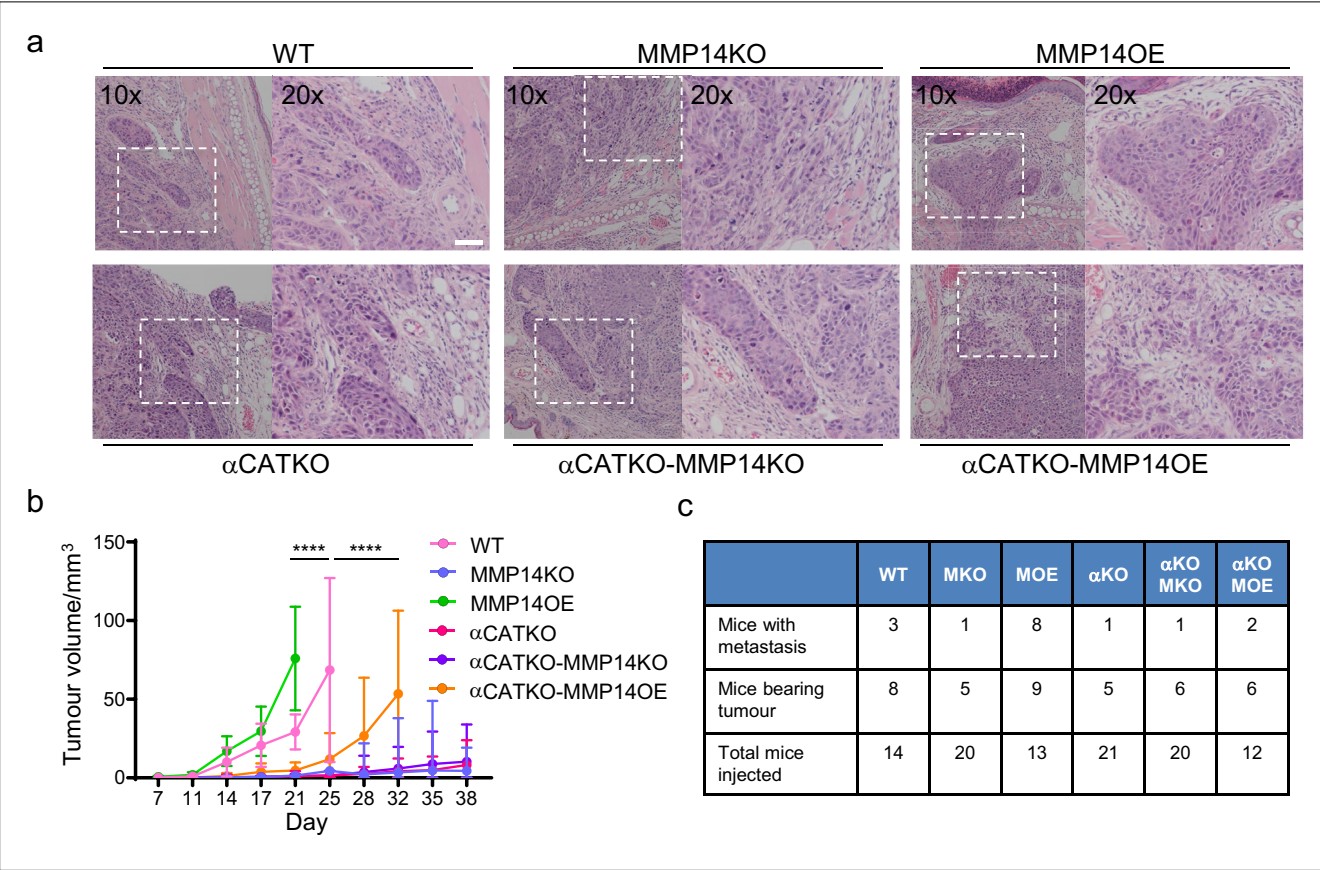

**Figure 9.** Strand widening is linked to tumour growth and metastasis. (**a**) H&E images are shown on tumours growing in the ears of mice with the indicated manipulations of MMP14 and CTNNA1. Scale bar = 50 μm. (**b**) Plot shows quantification of A431 tumour growth with the indicated manipulations of MMP14 and CTNNA1. (**c**) Table shows quantification of mice with primary tumours and mice with lymph node metastases when injected with A431 cells with the indicated manipulations of MMP14 and CTNNA1. The total number of mice for each condition also applies to the data plotted in (b). Two-way ANOVA with post-hoc multiple comparisons was performed. Error bars indicate 95% confidence intervals.

The online version of this article includes the following source data and figure supplement(s) for figure 9:

**Source data 1.** Tumour size and number of metastasis in WT and MMP14 and/or CTNNA1 manipulated tumour-bearing mice.

**Figure supplement 1.** Interplay of adherens junctions and matrix proteolysis determines the invasive pattern and growth.

**Figure supplement 1—source data 1.** Tumour invasion metrics.

over-expressing tumours grew and metastasised aggressively. This argues that SCC cells invading in thick strands are efficient at metastasis. Crucially, the aggressive behaviour of *MMP14* over-expressing cells is reduced by depletion of α-catenin. This argues strongly against a single cell form of migration being optimal for lymph node metastasis of SCC cells. The importance of adherens junctions for efficient metastasis is increasingly appreciated, this work suggests that one advantage of both adherens junctions and matrix proteases in collective migration is that the cells remain in a state capable of generating the space required for growth.

In silico analysis revealed that reducing cancer cell-ECM adhesion had a minor effect on determining the mode of invasion. Experimentation using *ITGB1* knock-out cells supported this analysis and notably confirmed the relationship between strand tapering and cancer cell-ECM adhesion. Moreover, unless cancer cell-ECM adhesion was very strong, the relationship between this variable and extent of invasion was rather weak in both organotypic and spheroid assays. Broadly, these data are consistent with the integrin independence of amoeboid forms of migration in 3D and hint at a role for either adhesion forces mediated by the glycocalyx or a role for outward forces that enable a 'chimneying' type of migration. In the future, it will be interesting to explore feedback loops between ECM properties, including density and stiffness, cell-matrix adhesion, cell behaviour, and low-affinity ECM adhesion mechanisms. A more sophisticated framework covering cell-ECM adhesion might also enable explanation of the experimental observation of thinner strands when ITGB1 is deleted (*Figure 4*). Cancer cell-cell adhesions exert a greater influence on collective cancer invasion than cell-ECM adhesions (*Figure 8—figure supplement 1d*). Intriguingly, the positive role of cancer cell-cell adhesions was most pronounced in simulations with a uniform chemotactic gradient. We propose that this reflects a crucial role of cell-cell adhesions in coordinating a supra-cellular actomyosin cytoskeleton in collectively invading clusters. This is likely to involve coordination of cell polarity complexes at sites of cell-cell contact. Interestingly, loss of cell-cell junctions was not sufficient to promote a clear switch to single cell invasion. This is likely due to the lack of available space in 3D contexts. This observation is consistent with *CDH1*-deficient tumours, such as invasive lobular carcinoma of the breast and some gastric cancers, typically showing thin strand-like patterns of invasion. Intriguingly, we observed transitions to single cell behaviour upon combined manipulation of CTNNA1 and MMP14 – either over-expression or knockout. The reasons for this are not immediately apparent, but the MMP14 knockout phenotype and the increase in single cell migration when ITGB1 and ITGB3 are depleted are consistent with protease- and adhesion-independent amoeboid cell migration (*Friedl and Wolf, 2010*; *Lämmermann et al., 2008*; *Tozluoğlu et al., 2013*; *Wolf et al., 2003*). Our modelling framework is not set up to consider amoeboid migration; hence, the transitions to single cell migration are not efficiently predicted. In future work, it will be interesting to integrate modelling frameworks for collective and amoeboid forms of migration.

To conclude, our integrated in silico and experimental approach reveals some of the key determinants of the mode of collective cancer invasion. Broad pushing fronts are associated with high matrix proteolysis and strong cancer cell-cell junctions and a lower dependence on CAFs. Reducing either proteolysis or cancer cell-cell adhesions leads to thinner invasive strands, with cell-matrix adhesions tuning strand tapering. We observe and experimentally demonstrate an unexpected linkage between the mechanisms that promote the widening of invasive strands and ability of cancer cells to grow when surrounded by ECM.

# Materials and methods

**Key resources table**

| Reagent type (species) or resource | Designation | Source or reference | Identifiers | Additional information |
|---|---|---|---|---|
| Antibody | Anti-MMP14 (rabbit monoclonal) | Abcam | ab51074 | WB (1:1000) |
| Antibody | Anti-alpha-catenin (rabbit monoclonal) | Abcam | ab51032 | WB (1:1000) |
| Antibody | Anti-vimentin (mouse monoclonal) | Sigma | SAB4200761 | WB (1:1000) |

*Continued on next page*

*Continued*

| Reagent type (species) or resource | Designation | Source or reference | Identifiers | Additional information |
|---|---|---|---|---|
| Antibody | Anti-fibronectin (rabbit polyclonal) | Sigma | F3648 | WB (1:1000) |
| Antibody | Anti-integrin β1 (mouse monoclonal) | Abcam | ab24693 | WB (1:1000) |
| Antibody | Anti-integrin β3 (rabbit monoclonal) | Abcam | ab179473 | WB (1:1000) IF (1:500) |
| Antibody | Anti-actin (mouse monoclonal) | Sigma | A4700 | WB (1:2000) |
| Antibody | Anti-pS19-MLC (rabbit polyclonal) | Cell Signaling | 3671 | IF (1:100) |
| Antibody | Anti-myosin MHC IIa (rabbit polyclonal) | Covance | PRB-440P | IF (1:100) |
| Antibody | Anti-β-catenin (mouse monoclonal) | Santa Cruz | sc7963 | IF (1:100) |
| Antibody | Anti-integrin β1 (mouse monoclonal) | Santa Cruz | sc13590 | IF (1:100) |
| Cell line (*Homo-sapiens*) | A431 | Cell Service department of Francis Crick Institute | | |
| Cell line (*Homo-sapiens*) | VCAF2B | Previously established (*Gaggioli et al., 2007*) | | |
| Transfected construct (*Homo-sapiens*) | px458 CTNNA1 gRNA | Santa Cruz | sc-419475 | |
| Transfected construct (*Homo-sapiens*) | px458 MMP14 gRNA | This paper | gctgctttgggccgagccg | Targeting gRNA sequence |
| Sequence-based reagent | siRNA: non-targeting | Dharmacon | siGENOME Non-Targeting Control siRNAD-001210-01-05 | Silencer Select Used at 20 nM |
| Sequence-based reagent | siRNA: targeting ITGB3 | Dharmacon | siGENOME SMARTpool M-004124-02-0010 | Silencer Select Used at 20 nM |
| Transfected construct (*Homo-sapiens*) | pMMP14-mCherry | Generous gift from Dr. Machesky at CRUK Beatson Institute | | |
| Transfected construct (*Homo-sapiens*) | pCSII-mCherry-CAAX | Previously generated | | Lentiviral construct to transfect and express membrane targeting mCherry |
| Transfected construct (*Homo-sapiens*) | pCSII-ECFP-CAAX | Previously generated | | Lentiviral construct to transfect and express membrane targeting ECFP |
| Transfected construct (*Homo-sapiens*) | pCSII-KEIMA-CAAX | Previously generated | | Lentiviral construct to transfect and express membrane targeting KEIMA |
| Chemical compound, drug | Collagen I | BD Biosciences | 354236 | |
| Chemical compound, drug | Matrigel | BD Biosciences | 354234 | |
| Chemical compound, drug | DQ collagen, type I from bovine skin, fluorescein conjugate | Thermo Fisher Scientific | D12060 | |
| Chemical compound, drug | Y27632 | Tocris Bioscience | 1254 | |
| Chemical compound, drug | 4-Hydroxytamoxifen (4OHT) | Sigma | H7904 | |
| Commercial assay or kit | Edu Cell Proliferation kit for imaging, Alexa Fluor 488 dye | Fisher Scientific | C10337 | |

## Experimental

### Cell culture

Human vulval CAFs are described in *Gaggioli et al., 2007*. CAFs were cultured in Dulbecco's Modified Eagle Medium (DMEM) supplemented with 10% Fetal Bovine Serum (FBS) and 1% insulin–transferrin–selenium (Invitrogen, no. 41400–045) and 100 U/ml penicillin, and 100 µg/ml streptomycin. Human vulval SCC cell line A431 cells were grown in DMEM supplemented with 10% FBS, 100 U/ml penicillin, and 100 µg/ml streptomycin. For ROCK inhibitor treatment cells were treated with 10 µM Y27632.

### Stable cell lines

*CTNNA1* or *MMP14* KO A431 cells were generated by CRISPR-Cas9 as previously described (*Labernadie et al., 2017*). Briefly, pX458 vectors encoding gRNA sequences were transfected into A431 cells, and single GFP positive cells were sorted into 96-well plate 2 days after transfection. Cells were grown for 2 weeks, and KO was checked by western blot and sequencing of genome DNA. For *MMP14* overexpressing cells, A431 cells were transfected with pMMP14-mCherry (generous gift from Dr. Machesky at CRUK Beatson Institute) and selected by G418 for 2 weeks. mCherry positive cells were sorted by flow cytometry. Stably labelled A431 cells and CAFs were obtained by infecting lentivirus containing fluorescent protein gene. 293 FT cells were transfected with pCSII-mCherry-CAAX, pCSII-ECFP-CAAX, or pCSII-KEIMA-CAAX construct and lentiviral RRE, REV, and VSVG encoding plasmids (5 µg each) by Xtremegene HP (Roche) according to the manufacturer's recommendation. Resulting supernatant containing lentivirus was then infected to target cells.

### Western blotting

Cells were lysed with Laemmli sample buffer containing 2.5% β-mercaptoethanol and heated at 95°C for 5 min. Samples were loaded to 4–15% polyacrylamide gels (Bio-Rad) for electrophoresis. Proteins were then transferred to a Poly Vinylidene DiFluoride (PVDF) membrane (Merck), which was blocked with 5% dry milk, Tris buffered saline, 0.2% Tween, and incubated with primary antibodies (overnight at 4°C) followed by secondary antibodies (1:10000) for 1 hr at room temperature. Proteins were detected by using Luminata Crescendo (Merck) and LAS600 (GE Healthcare). The following antibodies were used: anti-MMP14 rabbit monoclonal (1:1000, EP1264Y, Abcam), anti-alpha-catenin rabbit monoclonal (1:1000, EP1793Y, Abcam), anti-Vimentin mouse monoclonal (1:1000, 1A4, Sigma), anti-Fibronectin rabbit polyclonal (1:1000, Sigma), anti-integrin β1 mouse monoclonal (1:1000, P5D2, Abcam), anti-integrin β3 rabbit monoclonal (1:1000, ERP17507, Abcam), and anti-actin mouse monoclonal antibody (1:2000, AC-40, Sigma).

### Explant invasion assay

Human head and neck squamous cell carcinoma were collected with informed consent from all subjects and following ethical approval from the Institute of Cancer Research/Royal Marsden Hospital – reference CCR 2924. Frozen sections were stained as described previously (*Hidalgo-Carcedo et al., 2011 Calvo et al., 2013*). Patient-derived SCC tissues were chopped into small pieces (roughly 1 mm$^3$) and embedded in Collagen I/ Matrigel. Time-lapse images were taken by microscope every 10 min.

### Spheroid invasion assay

A431 and CAF cells were detached from the cell culture dishes with trypsin and re-suspended in sterile 0.25% methylcellulose solution in DMEM. The cellulose solution contained a 1:1 ratio of A431 and CAF cells at a concentration of $1 \times 10^5$ cells/ml. Twenty microlitre droplets were plated onto the underside of a 10-cm culture dish and allowed to form spheroids in a 37°C incubator overnight (hanging drop method). The spheroids were then embedded in a collagen I/Matrigel gel mix at a concentration of approximately 4 mg/ml collagen I and 2 mg/ml Matrigel (BD Bioscience) in 24-well glass-bottomed cell culture plates (MatTek) on a 37°C hot block. The gel was incubated for at least 30 min at 37°C with 5% $CO_2$. The gel was covered with DMEM media containing 10% FCS. Sixty hours later, the spheroids embedded in the gel were washed with PBS and then fixed for 30 min at room temperature with 4% paraformaldehyde. The spheroids were then imaged with an inverted Zeiss LSM780 at a magnification of ×10, ×20, and ×63. Z-stack images spanning 100–150 µm were collected, and image stacks were processed by ZEN software (Carl Zeiss) to yield maximum-intensity projections.

For quantification of the images, strand length and width were measured using Fiji software. Strand tapering was calculated by the following formula: strand width at 20% from the root/strand width at 80% from the root.

For EdU labelling experiment, spheroids were incubated with EdU containing medium for 1 hr prior to fix the samples.

For mitomycin C treatment experiment, cells were treated with 0.5 µg/ml mitomycin C for 24 hr prior to be subjected to hanging drop procedure.

## Organotypic invasion assay

Organotypic invasion assays were performed as previously described (*Gifford and Itoh, 2019*). Briefly, collagen I (BD Biosciences cat. No. 354249) and Matrigel (BD Biosciences cat. No. 354234) were mixed to yield a final collagen concentration of 4 mg/ml and a final Matrigel concentration of 2 mg/ml. After the gel had been left to set at 37°C for 1 hr, mixture of $5 \times 10^5$ A431 cells and $5 \times 10^5$ vulval CAFs (VCAFs) were plated on the top in complete medium. Twenty-four hours later, the gel was then mounted on a metal bridge and fed from underneath with complete medium (changed daily). After 6 days, the cultures were fixed with 4% PFA plus 0.25% glutaraldehyde in PBS and imaged using Zeiss LSM780 at a magnification of ×10 and ×20. Z-stack images spanning 100–150 µm were collected, and image stacks were processed by ZEN software to yield maximum-intensity projections.

For organotypic killing assay, the gels containing $5 \times 10^5$ VCAFs were set without cancer cells and incubated for 5 days in complete media. Then the gels were incubated with the media with puromycin (5 µg ml$^{-1}$) for 48 hr to kill the fibroblasts and then washed three times with complete media (30 min per wash). $5 \times 10^5$ cancer cells were then plated on top, and the assays proceeded as usual.

For quantification of the images, strand length and width were measured using Fiji software. Strand tapering was calculated by the following formula: strand width at 20% from the root/strand width at 80% from the root.

## Wound healing assay

$4 \times 10^4$ cells in 70 µL medium were seeded into each well of two-well culture insert (ibidi) and cultured overnight. After removing culture insert complete medium was added to the dish, and images were taken at 0, 9, and 24 hr. Empty area was measured using Fiji, and the results of 9 and 24 hr were normalised to that of 0 hr.

## Proliferation assays

2D assay – $5 \times 10^4$ cells were seeded in 24-well plate, and the number of cells was counted everyday using Countess II automated cell counter (Thermo Fisher Scientific). Results were normalised to day 1. 3D 'confined' assay – SCC cells were mixed in collagen I/Matrigel at a concentration of $3 \times 10^3$ / ml, and 100 µL of the mixture was put in 96-well plate and incubated for an hour at 37°. After the incubation, 150 µL of complete medium was added to each well. Images of growing cells were taken at indicated time points with EVOS FL microscope system (Thermo Fisher Scientific).

## ECM adhesion assay

Six-well plate was coated with collagen I (20 µg/ml) and Matrigel (20 µg/ml) for 2 hr. Cells were detached with Cell Dissociation Buffer enzyme-free (GIBCO), and $1 \times 10^5$ cells were seeded in each well. After 15 min of incubation, wells were washed twice with PBS, and cells were fixed with PFA. The number of cells in each field of view was counted to quantify the ECM adhesion ability of the cells.

## Collagen and collagen proteolysis imaging

Cells were seeded in a collagen/Matrigel mix as described for the proliferation assays. Collagen fibres were imaged using reflectance imaging on a confocal microscope. For timelapse analysis, cell cultures were maintained at 37°C and 5% $CO_2$. To visualise collagen proteolysis, the collagen/Matrigel mix was supplemented with 50 µg/ml DQ Collagen I. Collagen proteolysis was then imaged using a confocal with excitation at 488 nm and emission in the range 490–540 nm.

## Immunostaining

Cells were fixed with 4% paraformaldehyde for 10 min and permeabilised in 0.1% Triton X-100 for 10 min. Cells were blocked in 1% BSA for 1 hr before incubation with primary antibodies – pS19-MLC (Cell Signaling #3671 L), myosin MHC IIa (Covance PRB-440P), fibronectin (Sigma F3648), β-catenin (Santa Cruz sc7963), integrin β1 (Santa Cruz sc13590), and integrin β3 (Abcam, ab179473) at 4°C overnight. After incubation, the appropriate fluorescence-conjugated secondary antibodies for 1 hr, cells were washed with PBS. Images were acquired with an inverted Zeiss LSM780 at a magnification of ×20 and ×63. For quantification of the pMLC staining, regions of interest were drawn around equal numbers of 'free boundary zones' of A431 cells in clusters and cell-cell contact zones, and the mean fluorescent intensity was measured. The values were then normalised to the mean of all the boundary and contact zones for WT A431 cells. Staining of frozen human tissue sections was performed in a similar manner, except that fixation and permeabilisation times were doubled, and 5% BSA was used as a block.

## In vivo tumour growth

Cells were detached from culture flask and resuspended in 4 mg/ml Matrigel/PBS at a concentration of $2.5 \times 10^7$. Twenty microlitre of cell suspension was injected into ear intradermis of athymic nude mouse using 31 G needle (BD). The tumour size was measured every 3–4 days using caliper until it reached 0.6 mm in diameter. At the end point, mice were sacrificed, and the tumour samples were fixed with 4% PFA overnight and processed by standard methods for haematoxylin and eosin staining. Cervical lymph node was taken out and analysed for metastatic seeding.

## Computational

### Cellular Potts model

Detailed information on mathematical background and C++ coding implementation for each cellular mechanism within the model can be found in Appendix 1 in *Supplementary file 1* and at the GitHub repository https://github.com/RobertPJenkins/kato_jenkins_et_al_CC3D, (copy archived at swh:1:rev:b730d817f5c9cb11a4b3c5e02ccf03c829395fff; *Jenkins, 2022*).

### Simulation quantification

MATLAB functions quantifying invasion metrics can be found at the GitHub repository listed above. For each simulation outcome of interest (e.g. each combination of parameter values), 10 CC3D simulations were run to generate invasion metrics. All invasion metrics were calculated in MATLAB (version 2019a). Unless stated, all invasion metrics were recorded at day 4. Invading cells are classed as all cells beyond the tumour interface at day 0. Maximum invasion is given by the maximum distance of any invasive SCC centroid to the initial tumour interface. Invasion score is equal to the total number of invasive SCCs multiplied by the mean distance of invasive centroids to the initial tumour interface. For mean number of SCC neighbours, tapering and number of fractured objects in the bulk tumour mass at day 4 are found. The mean number of SCC neighbours is calculated for all cells in the bulk tumour mass that is invading. For these cells, the gradient of line of best fit between the number of neighbours and distance from initial tumour interface is calculated to give the tapering metric. Fractured objects are defined as objects unconnected to the bulk tumour mass and containing at least one SCC. The number of these distinct objects is counted for the fractured object metric. For cell growth, the total number of SCCs versus time is recorded, and an exponential fitted to the resulting curve. For the combination of very large SCC-degradation (eight WT), SCC-SCC adhesion (two WT), and SCC-ECM adhesion (four WT) in the presence of CAFs, spheroids can become hollow and break apart. In such circumstances, there is no bulk tumour mass resulting in a mean number of neighbours of zero for the main tumour mass and a large number of fractured objects. There are four instances of this, and these data have been removed (leaving six simulations for this region of parameter space) prior to PCA analysis and heatmap generation. The track invasion score is taken at day 5. It is calculated by finding all points around a permissive track, beyond the initial tumour boundary where the ECM density is 0.75 or below (initial condition set to 1). These points are then weighted according to their distance from the boundary and then summed. For the spheroid permissive track simulations, both sides of the initial tumour mass are quantified. Track width is calculated as the maximum width of the invading

strand. Strands that are either non-invasive or where the entire tumour mass has invaded uniformly do not record track width values. Curvature is quantified on day 7. The leading invasive edge is reduced to two one-dimensional signals in x-z and y-z for mid-points of y and x, respectively. Each one-dimensional signal is then smoothed with a smoothing window of 50 pixels. The LineCurvature2D function (Dirk-Jan Kroon (2021). 2D Line Curvature and Normals, MATLAB Central File Exchange. Retrieved 9 November 2021. https://www.mathworks.com/matlabcentral/fileexchange/32696-2d-line-curvature-and-normals) is used to calculate curvature for each signal and the average taken. For all heatmaps, for each box the x-axis represents the percentiles from 0.5 to 99.5 (left to right) of all 10 simulations for that outcome of interest.

## Acknowledgements

We thank the Francis Crick Institute Advanced Light Microscopy facility, Cell Services, Flow Cytometry, the Biological Research facility and Experimental Histopathology facility for scientific and technical support. We thank Dr Laura Machesky for kindly providing an *MMP14* plasmid. We are grateful to lab members, past and present, for help and advice throughout this work. The work was funded by the Francis Crick Institute which receives its core funding from Cancer Research UK (FC001144, FC001003), the UK Medical Research Council (FC001144, FC001003), and the Wellcome Trust (FC001144, FC001003). R.P.J., H.J., X.F., and E.S. were additionally supported by ERC Advanced Grant CAN_ORGANISE, Grant agreement number 101019366. T. K. was supported by JSPS Kakenhi grant number JP19K21262, Marie-Curie action (HeteroCancerInvasion no. 708651), The Uehara Memorial Foundation and Kitasato University Research Grant for Young Researchers.

## Additional information

### Competing interests

Stefanie Derzsi: Stefanie Derzsi is affiliated with Hoffman La-Roche. The author has no financial interests to declare. The other authors declare that no competing interests exist.

### Funding

| Funder | Grant reference number | Author |
| --- | --- | --- |
| Japan Society for the Promotion of Science | JP19K21262 | Takuya Kato |
| Marie Curie | 708651 | Takuya Kato |
| Cancer Research UK | FC001144 | Takuya Kato |
| Wellcome Trust | FC001144 | Takuya Kato |
| Medical Research Council | FC001144 | Takuya Kato |
| Cancer Research UK | FC001003 | Melda Tozluoglu |
| Wellcome Trust | FC001003 | Melda Tozluoglu |
| Medical Research Council | FC001003 | Melda Tozluoglu |
| Uehara Memorial Foundation | | Takuya Kato |
| Francis Crick Institute | FC001144 | Takuya Kato<br>Robert P Jenkins<br>Stefanie Derzsi<br>Melda Tozluoglu<br>Antonio Rullan<br>Steven Hooper<br>Raphaël AG Chaleil<br>Holly Joyce<br>Xiao Fu<br>Paul A Bates<br>Erik Sahai |

| Funder | Grant reference number | Author |
| --- | --- | --- |
| Kitasato University | | Takuya Kato |
| European Research Council | ERC Advanced Grant CAN_ORGANISE, Grant agreement number 101019366 | Erik Sahai<br>Holly Joyce<br>Robert P Jenkins<br>Xiao Fu |

The funders had no role in study design, data collection and interpretation, or the decision to submit the work for publication. For the purpose of Open Access, the authors have applied a CC BY public copyright license to any Author Accepted Manuscript version arising from this submission.

## Author contributions
Takuya Kato, Formal analysis, Funding acquisition, Investigation, Visualization, Methodology, Writing – original draft, Writing – review and editing; Robert P Jenkins, Conceptualization, Resources, Data curation, Software, Formal analysis, Investigation, Visualization, Methodology, Writing – original draft, Writing – review and editing; Stefanie Derzsi, Antonio Rullan, Formal analysis, Investigation; Melda Tozluoglu, Software, Formal analysis, Investigation, Methodology; Steven Hooper, Holly Joyce, Investigation, Methodology; Raphaël AG Chaleil, Resources, Software; Xiao Fu, Methodology; Selvam Thavaraj, Resources; Paul A Bates, Resources, Methodology; Erik Sahai, Conceptualization, Resources, Formal analysis, Supervision, Funding acquisition, Visualization, Methodology, Writing – original draft, Project administration, Writing – review and editing

## Author ORCIDs
Takuya Kato ⓘ http://orcid.org/0000-0002-4972-657X
Robert P Jenkins ⓘ http://orcid.org/0000-0002-6186-7746
Selvam Thavaraj ⓘ http://orcid.org/0000-0001-5720-7422
Paul A Bates ⓘ http://orcid.org/0000-0003-0621-0925
Erik Sahai ⓘ http://orcid.org/0000-0002-3932-5086

## Ethics
The Francis Crick's Institute Animal Welfare and Ethical Review Body and UK Home Office authority provided by Project License PP0736231 approved all animal model procedures. Every procedure of this study is compliant with all relevant ethical regulations regarding animal research.

## Decision letter and Author response
Decision letter https://doi.org/10.7554/eLife.76520.sa1
Author response https://doi.org/10.7554/eLife.76520.sa2

# Additional files

## Supplementary files
• Supplementary file 1. Appendices. Appendix 1: A walkthrough of developed CC3D plug-ins and steppables. Appendix 2: Additional CC3D parameter values. Appendix 3: PCA loadings and variance explained.

• Transparent reporting form

## Data availability
Source data for experiments is included in source data spreadsheets. Source code for Cellular Potts model and MATLAB quantification hosted on GitHub and comprehensively described in supporting walkthrough document (Appendix 1). Cellular Potts model parameter values are in Table 1 and Appendix 2.

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
