## [Editor Report]

This article constitutes a carefully written and highly impactful study that agrees with recent paradigm-shifting studies (e.g., doi: 10.1126/scitranslmed.abn7571 and doi: 10.1242/jcs.259275) suggesting that collective cell migration is the most efficacious way for epithelial cells to metastasize. The study uses mathematical modeling and experimental 3D approaches to demonstrate that cells necessitate space to both proliferate and invade as collective thick "pushing" strands. Importantly, extracellular matrix patterning provides uniform directional cues that harness adherens junctions and facilitate the collective thick strands of cells to 'push' and effectively travel through 3D microenvironmental settings. The study breaks new ground by incorporating cancer-associated fibroblasts and concludes that the pushing fronts are allied to extracellular matrix proteolysis and strong cancer cell-cell adherens junctions with diminished dependence on stromal cells (e.g., cancer-associated fibroblasts).

---

## [Decision Letter]

**Decision letter after peer review:**

Thank you for submitting your article "Interplay of adherens junctions and matrix proteolysis determines the invasive pattern and growth of squamous cell carcinoma" for consideration by *eLife*. Your article has been reviewed by 3 peer reviewers, and the evaluation has been overseen by a Reviewing Editor and Jonathan Cooper as the Senior Editor. The following individuals involved in the review of your submission have agreed to reveal their identity: Johanna Ivaska (Reviewer #1); Jaye C. Gardiner (Reviewer #3).

Essential revisions:

This study addresses several gaps that are evident with regards to cancer cell invasion in tissue. The approaches taken by this group encompassing mathematical modeling and experimental procedures are for the most part rigorous. The study, if some minimal remaining points are to be addressed, is deemed as highly impactful. The central points in need to be addressed are, (i) considering proliferation when interpreting invasion data, (ii) fully demonstrating cell-cell and cell-matrix dependent outcomes, (iii) considering ECM architecture, (iv) adding selected orthogonal approaches, and (v) addressing some predicaments concerning beta1-integrin deficient cells. Regarding the last listed point, in order to reach an assertive conclusion about beta1-integrin function and avoid the potential avb3-integrin compensation that is evident in beta1-null cells, beta1 loss needs to be rescued with WT vs the b1-integrin tryptophan mutant (doi: 10.1074/jbc.M300879200). Further, compensating experiments using specific integrin inhibitors and activator drugs (e.g. antibodies) might be of assistance to support the authors' interpretations.

*Reviewer #1 (Recommendations for the authors):*

This is a very interesting study addressing many aspects of the complicated regulation of cancer invasion in tissue. I am not an expert in computational modelling so I will focus my recommendation primarily on experimental validation.

1) The authors investigate many different parameters in their computational model. However, it seems that proliferation is not considered in the models. In particular, the organotypic assay is rather long (6 days) and strongly altered proliferation would be expected perhaps to influence the modeling outcome.

2) The authors conclude that proliferation is playing a key role in their system, however, proliferation is analysed with a methodology that is not matching with the invasion systems. It would be important to score cell proliferation in detail in the organotypic (for example by including Ki67 staining) and in the spheroid models. Where are the proliferating cells? Does the spatial distribution of the proliferation explain some of the patterns observed? Can this be modelled? On a related note, the fibroblasts employed in the organotypic assays are likely to secrete also ECM-tethered growth factors. Is the increased proliferation of the MMP14OE cells linked also to increased availability of stromal growth factors and not only to reduced confinement as suggested by the authors?

3) The authors should provide more detailed evidence on the disruption of the cell-cell adhesions/junctions in their aCatKO cells. Currently, these are lacking. What is the effect on b-catenin signaling. What happens to the cell-cell contacts? In Figure 6a they seem to be rather tightly clustered.

4) The authors have a nice structure in their study by addressing the contribution of different parameters one by one and providing modelling and experimental data to report the outcome. However, it is not clear why only some of the invasion parameters are provided for the experimental data in some cases. For example, in Figure 3 only strand length and width are provided, in Figure 4 only tapering and strand length and so on. For example in 4c it seems that strand width is rather different in the wt and b1KO cells.

5) Details of generating the b1KO cells are missing and there are no data provided validating their KO or addressing if there is compensation by other integrins. It is common in many cells types to have compensatory upregulation of ITGB3 upon b1KO. Similar to my comments related to the aCAT KO, the authors should demonstrate that the B1KO cells have reduced cell adhesion (to the matrixes used in their experimental setups) to validate their conclusion on adhesion strength and invasion.

6) would it be possible to provide quantification of the microscopy from several experiments to support the claims in Figure 6 a-c, which are currently based on representative micrographs or a limited number of cells?

*Reviewer #2 (Recommendations for the authors):*

For the experiments in Figure 4, a western blot confirming β1 integrin deletion is absent. The β1 integrin knockout invasive phenotypes are not intuitive, particularly the different modes of collective invasion across their two in vitro models and the apparent decrease in cell-cell adhesion (neighbors) in the organotypic model. A greater analysis and discussion of these phenotypes in the context of their proposed model would strengthen the work. The presence of Matrigel and other fibroblast-derived ECM could promote β1-independent cell-ECM adhesion strategies. A general perturbation analogous to deletion of α-catenin, such as deletion of talin, would more convincingly abrogate cell-ECM adhesion.

In line with the previous comment, there are several references to "pushing" or extensile forces driving the invasive growth of large collectives, supporting a diminished role for cell-ECM adhesion. These behaviors should be demonstrated in context, optically with either fiduciary markers (beads) or matrix displacements, to support key driving mechanisms.

The case for supracellular actomyosin coordination would be strengthened by quantitative immunostaining within invading collectives with different phenotypes (compared to the small cell islands in 2D/3D? experimental details are not clear in Figure 6).

In Figure 7, it is a bit difficult to appreciate from the images, but the authors should discuss how invasion in confined α-catenin knockout spheroids can be increased by loss or gain of MMP14 levels. This is at odds with predictions from the Potts model in Figure 7a.

In Figure 8b, proliferation should be decoupled from any differences in cell size, morphology, or organization that might affect spheroid morphology by immunostaining and quantifying proliferative markers.

The phenotypic results from the xenograft experiments are difficult to interpret and more description should be provided. For instance, are αCATKO, MMP14KO tumors invading with a wide front? Given the expected heterogeneity, phenotypes of tumor invasive fronts in Figure 8d should be quantified.

A summary schematic that contextualizes core findings and their specificity to each model should be added to aid comprehension.

*Reviewer #3 (Recommendations for the authors):*

When commenting on suggestions to increase the impact of the work, I have divided my comments into 3 sections: (1) suggested experiments/analyses, (2) points for consideration, and (3) questions for clarity. I hope the authors find my comments helpful.

i. Suggested experiments/analyses

As I have no expertise in computational modeling, I cannot fairly provide critiques or suggestions on the methodologies used. As such, all of my commentary is focused on the experimental aspects of this paper.

1. Other parameters that would be interesting to model/test in this system that have been noted to affect invasion elsewhere would be the stiffness of the ECM as well as its alignment. See Ahmadzadeh 2017 (PMID: 28196892) and Provenzano 2006 (PMID: 17190588). Ahmadzadeh et. al may be of interest to the authors all as it also includes some computational modeling in this topic.

2. It's interesting that both increasing and decreasing MMP14 cell levels affected invasion strand length. This may suggest that some ECM is necessary for invasion while too much can be inhibitory. While the authors largely focused on the effects on invasive strand width, I believe that there is some unique biology to uncover with regards to the strand length that the author's system is well poised to interrogate. Therefore, I believe that modeling matrix alignment and stiffness would be a great addition to this work.

3. There was a remarkable difference between the organotypic vs. spheroid models with regards to the effects of MMP14 manipulation. Later in the text the authors noted and explored the distribution of chemotactic signals as a potential difference. Could another possibility (though not mutually exclusive to the authors' hypothesis) be the placement of the fibroblasts underneath the cancer cells as opposed to intercalated within? Computation modeling would be sufficient for this question or at the very least be an interesting point to add to the discussion.

4. A useful control experiment would be to demonstrate that knockout and overexpression of MMP14 affects the amount of matrix present as expected. Indirect immunofluorescence of the collagen substrate used should be sufficient. This would not change the concept of the paper but further strengthen the authors' arguments.

5. Were the number of invasive strands different between wild type and ITGB1 KO in the organotypic model? From the representative image in Figure 4C it seems like there are more strands in the ITGB1 KO condition. While ITGB1 KO may have had no effect on strand length or tapering, it may have had an effect on other parameters. It would be useful for the authors to either include the graphs as supplementary data or to comment on it in the text.

6. The authors should include details about the ROCK:ER system (including how it was generated, why Tamoxifen (4OHT) was used, etc.) in the methods section.

7. Figure 6a – it would be helpful for the authors to include a nuclei stain (especially for the WT condition, but show it for both), to demonstrate that multiple cells are present but the pMLC is forming a supra-cellular structure as opposed to what's observed in the CTNNA1 KO samples. Would also suggest keeping the CTNNA1 KO nomenclature the same within the body of the figure and figure legend/text.

8. At the beginning of the text (line 161), the authors note fibroblast-matrix adhesions as one of the parameters to be tested but I am unclear of which set of experiments directly address this question. To my understanding, the ITGB1 KO experiments (and all genetic manipulations) were conducted in the A431 SCC cells as opposed to the fibroblasts. I suggest the authors either remove this parameter in the opening text or add text clarifying/specifying which experiments address this goal if the experiments were already performed.

ii. Points for consideration

1. Is ITGB1 highly expressed in normal tissue as compared to SCC? That is to say, is the high expression of ITGB1 in SCC biologically relevant to the tumor or the happenstance of a tumor forming in this cell type?

2. Are CAF-SCC heterotypic interactions necessary for migration?

3. Figure 1a box I – Could the broad "pushing" invasive masses of cells be a result of cells that break free from an invading strand and then grow separately?

4. While the authors found no consistent association between cancer cell and matrix adhesions (line 263 and figure 4a) could the importance of the matrix (as suggested by their MMP14 manipulation data) come from CAF-matrix adhesions? As the authors noted that invasion was boosted due to fibroblasts (line 210-211; sup. Figure 2f), this connection could have important ramifications on SCC ability to invade.

5. In Figure 6d, were fibroblasts present? It does not seem obvious from the images included. Additionally, though the authors were focusing on the effects of strand width and MMP14, there were also effects in strand length (Figure 3d). How does treatment with the ROCK inhibitor in MMP14 OE cells affect strand length?

6. Discussion of whether the in vitro organotypic or spheroid models better recapitulates what occurs in vivo would be beneficial.

iii. Questions for clarity

1. For a point of clarity – is "matrix displacement" referring to the alignment of the fibers? If yes, it would be beneficial to discuss this parameter in this context as alignment has been suggested to affect metastases and would help connect the authors' work to the broader literature. If no, it would be helpful for the authors to clarify what "displacement" means.

2. For clarity – are 'tapering' and 'strand width' similar measurements? If yes, I would suggest keeping the nomenclature the same between figures (figure 3d vs figure 4d). If no, it would be useful to clarify in the text what the difference in the measurement is and why the authors are focusing on this measurement over the other.

Recommendations for improving the writing and presentation.

On the whole, the text was very well written and easy to understand. The only section that could use some more clarity would be lines 238 to 241, but this may have been my difficulty with the associated figure as opposed to the text itself. Otherwise, my only other suggestion would be to keep the nomenclature of your cell lines consistent between the text, figure, and figure legend (for example, the α-catenin KO cells were described as αKO, αCATKO, and CTNNA1 KO).

---

## [Author Response]

Reviewer #1 (Recommendations for the authors):This is a very interesting study addressing many aspects of the complicated regulation of cancer invasion in tissue. I am not an expert in computational modelling so I will focus my recommendation primarily on experimental validation.

We are pleased that the reviewer describes our study as very interesting.

1) The authors investigate many different parameters in their computational model. However, it seems that proliferation is not considered in the models. In particular, the organotypic assay is rather long (6 days) and strongly altered proliferation would be expected perhaps to influence the modeling outcome.

The reviewer raises a good point. We have now reduced the proliferation rate by 50% in the model and find that the central findings of proteolysis and adherens junctions favouring wider strands are not altered. The data regarding proteolysis are now presented in Supp. Figure 3e, and the data regarding cancer cell – cancer cell adhesion are presented in Author response image 1. An additional, perhaps unsurprising, nuance to the analysis is that reduced proliferation is predicted to reduce strand width in spheroid assays – effectively, to have many neighbours you need to have cell proliferation. To test these predictions experimentally, we pre-treated cancer cells with mitomycin C to stop proliferation prior to setting up the spheroid assay. Supp. Figure 3h. Gratifyingly, this confirms the prediction that proteolysis increases strand width even when proliferation is absent. It additionally confirms that strands are narrower in the absence of proliferation.

**Author response image 1. sa2fig1:** Modelling the effect of low proliferation on invasion. Plots show the effect of reducing invasion and growth metrics (as determined by modelling) in organotypic conditions. Yellow indicates a high value and blue a low value. Note: the colour map has been normalised relative to the minimum and maximum values in each set of simulations to highlight the effect of varying SCC-SCC adhesion. Purple boxes highlight the similar relationship positive relationship between SCC – SCC adhesion and invasion regardless of proliferation rate.

2) The authors conclude that proliferation is playing a key role in their system, however, proliferation is analysed with a methodology that is not matching with the invasion systems. It would be important to score cell proliferation in detail in the organotypic (for example by including Ki67 staining) and in the spheroid models. Where are the proliferating cells? Does the spatial distribution of the proliferation explain some of the patterns observed? Can this be modelled? On a related note, the fibroblasts employed in the organotypic assays are likely to secrete also ECM-tethered growth factors. Is the increased proliferation of the MMP14OE cells linked also to increased availability of stromal growth factors and not only to reduced confinement as suggested by the authors?

As mentioned above, the issue of proliferation is important and interesting. We have addressed both computationally and experimentally.

As suggested by the reviewer, we have also explored the location of proliferating cells in our model (shown in blue in Author response image 2). In the example shown, we varied the proteolysis term in cancer cell only spheroids, which has a strong effect on growth (shown in Figure 8). As expected, more proliferating cells are observed when proteolysis is high. Moreover, they are fairly uniformly distributed.

**Author response image 2. sa2fig2:** Modelling the location of proliferated cells. Images show the location of cells that have proliferated in the timeframes indicated in blue (analogues to EdU labelling), red shows cells that have not proliferated. Upper panels show a spheroid with no ECM proteolysis, lower panels show a spheroid with high proteolysis.

We have also performed experimental analysis of the location of proliferating cells. Ki67 staining unexpectedly labelled almost all cells; therefore, we turned to EdU labelling, which provides a more definitive labelling of cells replicating their DNA. This reveals that proliferating cells are observed in both invading and non-invading cells, which is consistent with the modelling. These new data are shown in Supp. Figure 8a. Quantification of proliferating cells in invading strands further supports the association between strand widening. These new data are shown in Supp. Figure 8b. To address the issue of whether proliferation is a cause or consequence of invasive pattern we have performed new experiments using mitomycin to block the proliferation of cancer cells (Supp. Figure 3h). This reveals that proteolysis still influences invasive strand width in the absence of cancer cell proliferation. Crucially, this indicates that differences in proliferation are a consequence of invasive pattern.

Given the already extensive scope of the manuscript and its length, we feel that including a thorough consideration of the location of proliferating cells in both the experiments and modelling is beyond the scope of the work, and have prioritised presenting the experimental analysis. Nonetheless, we thank the reviewer for this suggestion and hope that they are suitably re-assured about the consistency between the model and experiments.

The question of possible growth factor processing by MMP14 over-expression is very interesting. Our data argue that growth factor processing cannot be the dominant mechanism at play because CTNNA1 KO reduces the effect of MMP14 over-expression when it would not be expected to have any effect of growth factor processing.

3) The authors should provide more detailed evidence on the disruption of the cell-cell adhesions/junctions in their aCatKO cells. Currently, these are lacking. What is the effect on b-catenin signaling. What happens to the cell-cell contacts? In Figure 6a they seem to be rather tightly clustered.

We now provide analysis of adherens junctions and β-catenin/CTNNB1. CTNNA1 KO leads to a clear loss of β-catenin staining at sites of cell-cell contact (Figure 6a), with only small puncta remaining. The reviewer is correct that the cells do retain some tendency to cluster. However, we should also point out that for the images showing the disruption of supra-cellular actomyosin, we deliberately showed CTNNA1 cells that are somewhat clustered, to make that point that even if these cells do contact each other they are unable to coordinate their cytoskeleton. The staining also shows that loss of CTNNA1 does not lead to nuclear accumulation of β-catenin (Figure 6a). This argues against a Wnt-driven transcriptional response – presumably because CTNNA1 is not sufficient to promote the other factors required for β-catenin to drive transcription, such as inhibition of GSK3-mediated phosphorylation. We also present analysis of Wnt/β-catenin target genes and for a known β-catenin target gene – Axin2 – from an RNA sequencing experiment comparing WT and CTNNA1 KO cells. This shows little consistent difference in between WT and KO cells, with a roughly equal number of target genes up- and down-regulated and p-value of 0.22. Moreover, *AXIN2* mRNA levels are not changed. These data are consistent with the lack of nuclear accumulation observed by in the immunostaining.

**Author response image 3. sa2fig3:** Analysis WNT/β-catenin signalling αCAT KO cells. Left panels show the lack of significant gene enrichment score for WNT/β-catenin target genes in αCAT KO cells (lower yellow rows) compared to WT (upper grey rows). Right panel shows the levels of a AXIN2 mRNA in WT and αCAT/CTNNA1 KO cells. Data determined by RNA sequencing.

4) The authors have a nice structure in their study by addressing the contribution of different parameters one by one and providing modelling and experimental data to report the outcome. However, it is not clear why only some of the invasion parameters are provided for the experimental data in some cases. For example, in Figure 3 only strand length and width are provided, in Figure 4 only tapering and strand length and so on. For example in 4c it seems that strand width is rather different in the wt and b1KO cells.

We now provide the additional quantification in Figures 3d, 4d, and 5d. It was not included in the original submission as we were attempting (erroneously) to streamline the presentation for ease of reading. This quantification confirms the reviewer’s comment that ITGB1 KO cells have thinner strands. Interestingly, this is not predicted by our model and we do not have a definitive explanation for the result. A more sophisticated handling of cell-ECM interactions in future iterations of the model may address this. For example, we speculate that it might reflect a role for ITGB1 in cancer cell – cancer cell cohesion, possibly mediated by binding to ECM molecules located between cells. We and others have previously observed this in cancer cell models (Vial et al., Cancer Cell 2003 – Figure 4F). We now include a sentence in the discussion relating to the narrower strands in ITGB1 KO cells.

5) Details of generating the b1KO cells are missing and there are no data provided validating their KO or addressing if there is compensation by other integrins. It is common in many cells types to have compensatory upregulation of ITGB3 upon b1KO. Similar to my comments related to the aCAT KO, the authors should demonstrate that the B1KO cells have reduced cell adhesion (to the matrixes used in their experimental setups) to validate their conclusion on adhesion strength and invasion.

We apologise for this omission. We now include data showing that ITGB1 KO have a 90% reduction in their ability to bind to collagen I (Supp. Figure 4b). We also show that ITGB3 levels are not affected (Supp. Figure 4d and e). Moreover, combined targeting of ITGB1 and ITGB3 does not reduce invasion (Supp. Figure 4f).

6) would it be possible to provide quantification of the microscopy from several experiments to support the claims in Figure 6 a-c, which are currently based on representative micrographs or a limited number of cells?

The reviewer makes a good suggestion – similar to one made by reviewer #2. We have now included quantification of the relative level of pMLC at the edge of the cell cluster vs the sites of cell-cell contact. This provides numeric support for our assertion that loss of CTNNA1 or ectopic ROCK2 activation disrupts the supra-cellular actomyosin network. These new analyses are in Supp. Figure 6a.

Reviewer #2 (Recommendations for the authors):For the experiments in Figure 4, a western blot confirming β1 integrin deletion is absent. The β1 integrin knockout invasive phenotypes are not intuitive, particularly the different modes of collective invasion across their two in vitro models and the apparent decrease in cell-cell adhesion (neighbors) in the organotypic model. A greater analysis and discussion of these phenotypes in the context of their proposed model would strengthen the work. The presence of Matrigel and other fibroblast-derived ECM could promote β1-independent cell-ECM adhesion strategies. A general perturbation analogous to deletion of α-catenin, such as deletion of talin, would more convincingly abrogate cell-ECM adhesion.

The reviewer makes a valid point. We now include data showing that ITGB1 KO have a 90% reduction in their ability to bind to collagen I (Supp. Figure 4c). We also show that ITGB3 levels are not affected. Moreover, combined targeting of ITGB1 and ITGB3 does not reduce invasion (Supp. Figure 4d-f) As suggested by the reviewer, we attempted talin depletion but found that either depletion was not very effective or detrimental to cell viability (data not shown). Given that ITGB1 KO alone reduces cell-ECM adhesion by ~90%, we are confident that combined ITGB1 and ITGB3 targeting will lead to an almost complete block of cell-ECM adhesion in our assays. Interestingly, we note a transition to single cell invasion when ITGB1 and ITGB3 deletion/depletion are combined. This is potentially consistent with the integrin-independence of amoeboid cancer cell migration (now discussed on page 16, lines 525-528). We now also include discussion of the caveat that we cannot exclude integrin-independent adhesion mechanisms. Nonetheless, we would like to re-iterate that the main emphasis and conclusions of this study relate to the interplay of proteolysis and adherens junctions.

In line with the previous comment, there are several references to "pushing" or extensile forces driving the invasive growth of large collectives, supporting a diminished role for cell-ECM adhesion. These behaviors should be demonstrated in context, optically with either fiduciary markers (beads) or matrix displacements, to support key driving mechanisms.

We thank the reviewer for this excellent, and somewhat challenging, suggestion. The measurement of pushing/compressive forces in 3D is not a ‘routine’ technique. Nonetheless, after some optimisation, we have now imaged the collagen surrounding small clusters of cancer cells with different junctional and protease functionality. These new data are presented in Figure 8. In brief, we observe some compaction of collagen fibres surrounding the clusters of control A431 cells. This is not observed in CTNNA1 KO cells. In addition, MMP14 over-expression also alters the interface between the cell cluster and the collagen fibres. The compaction of collagen is lost when MMP14 expression is elevated and there is additionally a reduction in membrane blebs at the interface – this later point was already visible in the original phase contrast images in Figure 8b. The loss of compaction can be explained by the fact that the collagen is removed via degradation in this context, and not pushing forces. Furthermore, several studies have established that membrane blebbing is a response to compression from the surrounding environment. We therefore interpret the presence of blebbing in the control and CTNNA1 KO to indicate that the clusters are compressed, but that matrix proteolysis can relieve this compression.

The case for supracellular actomyosin coordination would be strengthened by quantitative immunostaining within invading collectives with different phenotypes (compared to the small cell islands in 2D/3D? experimental details are not clear in Figure 6).

The reviewer makes a good suggestion – similar to one made by reviewer #1. We have now included quantification of the relative level of pMLC at the edge of the cell cluster vs the sites of cell-cell contact. This provides numeric support for our assertion that loss of CTNNA1 or ectopic ROCK2 activation disrupts the supra-cellular actomyosin network. These new analyses are in Supp. Figure 6c.

Regarding evidence for the supracellular actomyosin coordination in 3D contexts: we have added staining of human SCC that shows supracellular coordination of MYH9/MHCIIA (Supp. Figure 6d). We have also shown supracellular coordination of MYL9/MLC2 in 3D models of invasion Gaggioli et al., (Nature Cell Biology 2007).

In Figure 7, it is a bit difficult to appreciate from the images, but the authors should discuss how invasion in confined α-catenin knockout spheroids can be increased by loss or gain of MMP14 levels. This is at odds with predictions from the Potts model in Figure 7a.

The reviewer makes an astute point: the model does not predict that CTNNA1 KO will increase the invasion distance when proteolysis is high. We do not have definitive explanation for this, but we believe it relates to the experiments starting to transition to single cell modes of invasion, which the Potts model is not set up for. Two observations support this: (1) we observe single invading cells in the MMP14 o.e. CTNNA1 KO setting (2) the Potts model predicts increased fragmentation in the context of high proteolysis. This is now discussed on page 16, lines 523-531.

In Figure 8b, proliferation should be decoupled from any differences in cell size, morphology, or organization that might affect spheroid morphology by immunostaining and quantifying proliferative markers.

The reviewer raises an interesting point about the relationship between size, morphology, organisation and proliferation. We respond in several ways. First, the magnitude of the effect of sphere volume by day 8 spans over one order of magnitude, which cannot be explained by some variation in cell size or how the spheres are packed. Second, we now investigate differences in cell size at day 4 of these assays (shown in Author response image 4), when the differences in sphere volume are small. This confirms that there are no differences in the average cross-sectional area of the cells or their nuclei. As documented in response to reviewer #2 point #2, there are some differences in the organisation of the cluster-matrix interface. However, the presence or absence of blebs that are typically 1-4 microns in size cannot not explain the differences in the volume of spheres that are up to 200 microns across. Lastly, as discussed in response to reviewer #1 point #1 we can exclude changes in proliferation being the cause of differences in invasive pattern through the use of mitomycin C.

**Author response image 4. sa2fig4:** Size of cells in 3D following perturbations. Plot shows the effect of the indicated manipulations on the cross-sectional area of A431 cells when grown in spheroid conditions mean and 95% confidence interval are plotted.

The phenotypic results from the xenograft experiments are difficult to interpret and more description should be provided. For instance, are αCATKO, MMP14KO tumors invading with a wide front? Given the expected heterogeneity, phenotypes of tumor invasive fronts in Figure 8d should be quantified.

We agree that quantification will strengthen the manuscript and now provide it in Supp. Figure 9b. These data show that the greatest invasion (as scored by the product of the distance invaded and the number of cells) is associated with high proteolysis – MMP14OE. Moreover, the cells clusters have a low aspect ratio (indicating that they are round). αCATKO leads to many fewer invading clusters, with lower invasion scores, and higher aspect ratios. Analysis of MMP14KO tumours was hampered by the rarity of invading cell clusters. Taken together, these data are consistent with the main themes of our study.

A summary schematic that contextualizes core findings and their specificity to each model should be added to aid comprehension.

We have updated our schematic to improve its comprehensibility – Supp. Figure 9c

Reviewer #3 (Recommendations for the authors):When commenting on suggestions to increase the impact of the work, I have divided my comments into 3 sections: (1) suggested experiments/analyses, (2) points for consideration, and (3) questions for clarity. I hope the authors find my comments helpful.i. Suggested experiments/analysesAs I have no expertise in computational modeling, I cannot fairly provide critiques or suggestions on the methodologies used. As such, all of my commentary is focused on the experimental aspects of this paper.1. Other parameters that would be interesting to model/test in this system that have been noted to affect invasion elsewhere would be the stiffness of the ECM as well as its alignment. See Ahmadzadeh 2017 (PMID: 28196892) and Provenzano 2006 (PMID: 17190588). Ahmadzadeh et. al may be of interest to the authors all as it also includes some computational modeling in this topic.

We thank the reviewer for raising these interesting points. In the revised manuscript, we now present analysis of the effect of alignment (Supp. Figure 3d). As implied by the simulations of cancer cells following a narrow track, invasion in the direction of aligned ECM tracks is favoured with matrix aligned with direction of invasion increasing the invasion score and maximum invasion in organotypic simulations compared to chessboard distributed fibres. This is also observed experimentally as shown in Author response image 5.

**Author response image 5. sa2fig5:** Cancer cell invasion aligns with matrix pattern. Images shown in the pattern of SCC spheroid (A431 cells in cyan) invasion/migration 96 hours after being placed on top of an aligned fibroblast-derived matrix (red). White arrows indicate the local matrix alignment.

Regarding stiffness: this is not explicitly modelled in our simulations and it is not possible to change the stiffness of our experimental matrices without also changing other factors such as matrix density or crosslinking. Nonetheless, matrix stiffness can be considered in terms of the resistance to the pushing force that cells generate. Thus, we can mimic stiff matrix by reducing the effect/magnitude of the pushing force in the model. This leads to shorter strands and lower overall invasion scores. In the context of the organotypic models, there is a slight increase in tapering (shown in Author response image 6). Our model lacks signalling or feedback from cell-ECM adhesions, thus it is not well-suited to studying some of the more interesting aspects of the interplay between matrix stiffness and cell behaviours. Although it would be fascinating to implement this, it would be best done using other modelling frameworks, such as, and represent a new project in its own right. This is now discussed on page 15, lines 509-513.

**Author response image 6. sa2fig6:** The effect of matrix definition invasion. Plots show the effect of skewing the amount of force needed to remodel matrix in the model. Note high ‘stiffness’, which is modelled by reducing the ability to remodel the matrix is to the left of the plots. Purple boxes highlight how the distance and overall extent of invasion decreases when the relative matrix stiffness is high.

2. It's interesting that both increasing and decreasing MMP14 cell levels affected invasion strand length. This may suggest that some ECM is necessary for invasion while too much can be inhibitory. While the authors largely focused on the effects on invasive strand width, I believe that there is some unique biology to uncover with regards to the strand length that the author's system is well poised to interrogate. Therefore, I believe that modeling matrix alignment and stiffness would be a great addition to this work.

We agree with the sentiment of this comment and now included consideration of matrix alignment in Supp. Figure 3d (discussed above).

3. There was a remarkable difference between the organotypic vs. spheroid models with regards to the effects of MMP14 manipulation. Later in the text the authors noted and explored the distribution of chemotactic signals as a potential difference. Could another possibility (though not mutually exclusive to the authors' hypothesis) be the placement of the fibroblasts underneath the cancer cells as opposed to intercalated within? Computation modeling would be sufficient for this question or at the very least be an interesting point to add to the discussion.

We thank the reviewer for this suggestion. We have now included fibroblasts mixed in with the cancer cells in the organotypic assays *in silico*. These data are now presented in Supp. Figure 5e. In brief, the requirement for proteolysis and cancer cell – cancer cell adhesion for maximal strand width is still observed if fibroblasts are mixed with the cancer cells. These analyses exclude different fibroblast starting positions being the critical difference between the organotypic and spheroid assays.

4. A useful control experiment would be to demonstrate that knockout and overexpression of MMP14 affects the amount of matrix present as expected. Indirect immunofluorescence of the collagen substrate used should be sufficient. This would not change the concept of the paper but further strengthen the authors' arguments.

We agree. We have now included experiments using DQ collagen I, which exhibits increased fluorescence upon cleavage. These are presented in Supp. Figure 3g and confirm that the MMP14 KO and over-expression have the expected consequences on collagen proteolysis. We have also added reflectance imaging of the collagen fibres to Supp. Figure 8.

5. Were the number of invasive strands different between wild type and ITGB1 KO in the organotypic model? From the representative image in Figure 4C it seems like there are more strands in the ITGB1 KO condition. While ITGB1 KO may have had no effect on strand length or tapering, it may have had an effect on other parameters. It would be useful for the authors to either include the graphs as supplementary data or to comment on it in the text.

We have now included additional quantification in Figure 4c – specifically strand width. As the reviewer perhaps alludes to, there is a difference in the strand width parameter in the ITGB1KO setting. Interestingly, this is not predicted by our model and we do not have a definitive explanation for the result.

6. The authors should include details about the ROCK:ER system (including how it was generated, why Tamoxifen (4OHT) was used, etc.) in the methods section.

We apologise for this omission and have included more information as requested (page 11, lines 355-356). In brief, the regulatory regions of ROCK2 are replaced with the hormone binding domain of the oestrogen receptor. This has the consequence of rendering ROCK2 responsive to oestrogen like compounds, such as 4 hydroxy-tamoxifen. We cite Croft et al., Cancer Research 2004 in which the ROCK:ER system was first characterised.

7. Figure 6a – it would be helpful for the authors to include a nuclei stain (especially for the WT condition, but show it for both), to demonstrate that multiple cells are present but the pMLC is forming a supra-cellular structure as opposed to what's observed in the CTNNA1 KO samples. Would also suggest keeping the CTNNA1 KO nomenclature the same within the body of the figure and figure legend/text.

We agree and now include a nuclear staining and have changed the nomenclature to CTNNA1 throughout Figures 5, 6, and 8.

8. At the beginning of the text (line 161), the authors note fibroblast-matrix adhesions as one of the parameters to be tested but I am unclear of which set of experiments directly address this question. To my understanding, the ITGB1 KO experiments (and all genetic manipulations) were conducted in the A431 SCC cells as opposed to the fibroblasts. I suggest the authors either remove this parameter in the opening text or add text clarifying/specifying which experiments address this goal if the experiments were already performed.

We apologise for the lack of clarity. Fibroblast-matrix adhesions are included as a necessary feature of the model – fibroblasts bind to the ECM. However, it is not one that we varying computationally or experimentally in this study as our focus in on cancer cell determinants of invasive pattern. In the course of developing the model, we did alter fibroblast-matrix adhesion. If this is reduced, then invasion is reduced, which is consistent with the data for integrin depletion in fibroblasts that we published previously (Gaggioli et al., 2007 and Hooper et al., 2010).

ii. Points for consideration1. Is ITGB1 highly expressed in normal tissue as compared to SCC? That is to say, is the high expression of ITGB1 in SCC biologically relevant to the tumor or the happenstance of a tumor forming in this cell type?

In normal mucosal tissue, ITGB1 expression is restricted to the proliferative basal layer of cells. In mucosal SCC it is expressed in the majority of cells and therefore the overall level of expression in cancerous tissue is higher. In addition, the expression in cancer cells is often higher than in the adjacent basal cells in non-transformed tissue. This is documented in Brockbank et al., BJC 2005. We should also point out that ITGB1 expression is higher in stromal fibroblasts.

2. Are CAF-SCC heterotypic interactions necessary for migration?

This is a good question. In our computational model, heterotypic interactions are important when cancer cell-matrix adhesion is low, especially in spheroids (shown in Author response image 7). In experimental analysis, we have previously reported that CAF-SCC interactions are required for the efficient invasion of spheroids (Labernadie et al., 2017). Thus, there is good concordance between experiments and modelling regarding the importance of cancer cell – fibroblast adhesion on invasion.

**Author response image 7. sa2fig7:** Modelling the effect of Cancer Cell – Fibroblast adhesion on invasion. Plots show the effect of varying cancer cell – fibroblast adhesion of invasion and growth metrics (as determined by modelling) in both organotypic and spheroid conditions. Yellow indicates a high value and blue a ow value. Purple box highlights the positive relationship between cancer cell – fibroblast adhesion and invasion, expecially in the context of spheroids.

3. Figure 1a box I – Could the broad "pushing" invasive masses of cells be a result of cells that break free from an invading strand and then grow separately?

This is a good question and perfectly plausible. Unfortunately, we have no way of determining this in patient tissue as we lack dynamic information. In our model systems, we do not observe single cell invasion followed by proliferation to generate an invading mass.

4. While the authors found no consistent association between cancer cell and matrix adhesions (line 263 and figure 4a) could the importance of the matrix (as suggested by their MMP14 manipulation data) come from CAF-matrix adhesions? As the authors noted that invasion was boosted due to fibroblasts (line 210-211; sup. Figure 2f), this connection could have important ramifications on SCC ability to invade.

The reviewer is correct that fibroblast-ECM interaction is important to boost invasion. We have previously reported this in Gaggioli et al., Nature Cell 2007 and Hooper et al., British Journal of Cancer 2010.

5. In Figure 6d, were fibroblasts present? It does not seem obvious from the images included. Additionally, though the authors were focusing on the effects of strand width and MMP14, there were also effects in strand length (Figure 3d). How does treatment with the ROCK inhibitor in MMP14 OE cells affect strand length?

The reviewer makes an astute observation. The experiment in Figure 6d is a slight variation on the normal organotypic assay. To avoid treating the fibroblasts with the ROCK inhibitor and therefore confounding the analysis (we previously published that ROCK inhibition prevents fibroblasts from promoting cancer invasion – Gaggioli et al., 2007), we used an experimental set up in which fibroblasts are allowed to remodel the ECM prior to the addition of cancer cells and in the absence of any drugs. The fibroblasts are then removed (using detergent and thoroughly washing), and then cancer cells are added with or without the ROCK inhibitor or 4OHT. Thus, fibroblasts are used to promote the invasion of cancer cells, but are not present at the end of the assay. This explains the lack of magenta signal. We have now edited the text on page 11, lines 363-365 to make this clear.

6. Discussion of whether the in vitro organotypic or spheroid models better recapitulates what occurs in vivo would be beneficial.

This is discussed on page 6, lines 147-151 – now highlighted in green text. In brief, our view is that the organotypic assay better mimics the first stages of invasion when the tissue is layer with underlying dermis. The spheroid assay is probably a better recapitulation of the situation when cancer cells have already invaded into the deeper dermis and surrounded by tissue on all sides.

iii. Questions for clarity1. For a point of clarity – is "matrix displacement" referring to the alignment of the fibers? If yes, it would be beneficial to discuss this parameter in this context as alignment has been suggested to affect metastases and would help connect the authors' work to the broader literature. If no, it would be helpful for the authors to clarify what "displacement" means.

Matrix displacement is the transfer of matrix from one voxel to another. For example, if a cancer cell reduces the matrix ‘concentration’ in a neighbouring voxel and there is a corresponding increase in an adjacent matrix voxel, then this is displacement. If a cancer cell reduces the matrix ‘concentration’ in a neighbouring voxel and there is no increase in surrounding voxels, then this is degradation. In the case of persistently moving cell, matrix displacement could generate a track lined with high matrix density, which would be similar to alignment, but displacement is not the same as alignment.

2. For clarity – are 'tapering' and 'strand width' similar measurements? If yes, I would suggest keeping the nomenclature the same between figures (figure 3d vs figure 4d). If no, it would be useful to clarify in the text what the difference in the measurement is and why the authors are focusing on this measurement over the other.

They are not the same. Strand width is the average width of the strand, whereas tapering captures if the width of the strand reduces as it extends away from the main tumour mass. We have clarified this in the text relating to Supp. Figure 2b on pages 6&7 that describes the metrics we use.